# Magnetic stimulation allows focal activation of the mouse cochlea

Jae-Ik Lee[1†], Richard Seist[2,3†], Stephen McInturff[2,4], Daniel J Lee[2,4], M Christian Brown[2,4], Konstantina M Stankovic[2,4,5*], Shelley Fried[1,6*]

[1]Department of Neurosurgery, Massachusetts General Hospital, Harvard Medical School, Boston, United States; [2]Department of Otolaryngology - Head and Neck Surgery, Massachusetts Eye and Ear, Harvard Medical School, Boston, United States; [3]Department of Otorhinolaryngology - Head and Neck Surgery, Paracelsus Medical University, Salzburg, Austria; [4]Program in Speech and Hearing Bioscience and Technology, Harvard Medical School, Boston, United States; [5]Department of Otolaryngology – Head and Neck Surgery, Stanford University School of Medicine, Stanford, United States; [6]Boston VA Medical Center, Boston, United States

*For correspondence:
kstankovic@stanford.edu (KMS);
fried.shelley@mgh.harvard.edu
(SF)

[†]These authors contributed
equally to this work

Competing interest: The authors
declare that no competing
interests exist.

Reviewing Editor: Brice
Bathellier, CNRS, France

**Abstract** Cochlear implants (CIs) provide sound and speech sensations for patients with severe to profound hearing loss by electrically stimulating the auditory nerve. While most CI users achieve some degree of open set word recognition under quiet conditions, hearing that utilizes complex neural coding (e.g., appreciating music) has proved elusive, probably because of the inability of CIs to create narrow regions of spectral activation. Several novel approaches have recently shown promise for improving spatial selectivity, but substantial design differences from conventional CIs will necessitate much additional safety and efficacy testing before clinical viability is established. Outside the cochlea, magnetic stimulation from small coils (micro-coils) has been shown to confine activation more narrowly than that from conventional microelectrodes, raising the possibility that coil-based stimulation of the cochlea could improve the spectral resolution of CIs. To explore this, we delivered magnetic stimulation from micro-coils to multiple locations of the cochlea and measured the spread of activation utilizing a multielectrode array inserted into the inferior colliculus; responses to magnetic stimulation were compared to analogous experiments with conventional microelectrodes as well as to responses when presenting auditory monotones. Encouragingly, the extent of activation with micro-coils was ~60% narrower compared to electric stimulation and largely similar to the spread arising from acoustic stimulation. The dynamic range of coils was more than three times larger than that of electrodes, further supporting a smaller spread of activation. While much additional testing is required, these results support the notion that magnetic micro-coil CIs can produce a larger number of independent spectral channels and may therefore improve auditory outcomes. Further, because coil-based devices are structurally similar to existing CIs, fewer impediments to clinical translational are likely to arise.

## Editor's evaluation

This study demonstrates that microcoil magnetic stimulation in the cochlea generates efficient and spatially precise activation of the auditory system. This new method opens interesting possibilities to improve the resolution of auditory restoration by implanted devices.

## Introduction

More than 430 million people worldwide, ~5% of the world's population, live with disabling hearing loss, making it the most common sensory deficit (**WHO, 2021**). The World Health Organization (WHO) estimates that this number will grow to 700 million by 2050 (**WHO, 2021**). There are significant associations between hearing impairment and reduced quality of life, increased risk for dementia, and/or an inability to function independently (**Gurgel et al., 2014**). In the most common type of hearing loss, called sensorineural hearing loss, there is often a loss of the sensory hair cells that transduce sound-induced vibrations within the cochlea into neural activity; this loss precludes the use of sound-amplifying hearing aids as a potential treatment. Instead, a cochlear implant (CI) can be implanted to electrically stimulate spiral ganglion neurons (SGNs), the neurons downstream from hair cells. CIs are generally effective for enabling speech comprehension, with mean single syllable open set word recognition scores of 65% in quiet (**Firszt et al., 2004**; **Holden et al., 2013**; **Moberly et al., 2016a**, **Moberly et al., 2016b**). However, there is a significant reduction in CI performance when background noise levels are high and most CI users also cannot appreciate music (**Friesen et al., 2001**; **Jiam et al., 2017**). Thus, despite the unquestionable benefit provided by existing devices, there is room for improvement.

The limitations in performance are thought to arise largely from the small number of independent spectral channels created by CIs. In contrast to the large number of independent channels arising from the ~30,000 SGNs in the intact human cochlea, CIs produce as few as 8–10 independent spectral channels (**Friesen et al., 2001**; **Fishman et al., 1997**; **Fu and Nogaki, 2005**; **Carlyon et al., 2007**). This is smaller than the number of stimulating electrodes in most existing devices (**Fishman et al., 1997**; **Friesen et al., 2001**). The discrepancy is thought to arise from several intrinsic limitations associated with electric stimulation of the cochlea. For example, the high conductivity of the perilymph within the scala tympani leads to an expansive spread of current in the longitudinal direction (along the tonotopic axis; **Figure 1B**). In addition, the high electrical resistance of the bony wall separating the scala tympani from targeted SGNs within the organ of Corti necessitates an increase in the amplitude of stimulation that results in an even wider spread of activation. Excessive spread from individual electrodes can result in overlap of fields from neighboring electrodes, thereby reducing spectral specificity.

Several novel approaches are under consideration to increase the number of independent channels created by CIs. For example, the genetic insertion of light-sensitive ion channels into SGNs allows their activation to be controlled by light instead of electric fields, resulting in narrower channels (**Hernandez et al., 2014** ; **Dieter et al., 2019**). Another approach uses electrode arrays that penetrate directly into the auditory nerve (**Middlebrooks and Snyder, 2007**; **Middlebrooks and Snyder, 2008**; **Middlebrooks and Snyder, 2010**); the reduced separation between electrodes and targeted neurons, along with the elimination of the high-resistance barrier, enables better control of activation of the central axons of SGNs. While considerable progress has been made with both approaches, the use of genetic manipulations and/or new surgical techniques raise a number of important safety concerns that will need to be addressed prior to large-scale clinical implementation.

Recent studies have shown that magnetic stimulation from small, implantable coils, referred to as micro-coils, can effectively drive neurons of the central nervous system (CNS) while confining activation to a narrow region around each coil (**Lee et al., 2016**; **Lee and Fried, 2017**; **Lee et al., 2019**; **Ryu et al., 2020**). A coil-based approach is potentially attractive for CIs because magnetic fields are highly permeable to most biological materials, for example, the bony walls of the scala tympani, and thus activation of SGN processes would not require the same increase in stimulation amplitude required for electric stimulation. Further, the spread of magnetic fields in the scala tympani is less sensitive to the high conductivity of perilymph, further helping to confine activation (**Figure 1A**). While the strength of the fields induced by micro-coils is small, computational studies suggest that the spatial gradient of the resulting fields is suprathreshold (**Lee et al., 2016**; **Lee and Fried, 2017**; **Lee et al., 2019**; **Osanai et al., 2018**; **Minusa et al., 2018**; **Ryu et al., 2020**), and simulations specific to the cochlea suggest a multiturn spiral coil should produce fields strong enough to activate SGNs (**Mukesh et al., 2017**). It is not clear however whether spiral coil designs are best for use in a high-count, multicoil array designed for the cochlea, as they might reduce the flexibility of the implant and thus could increase the risk for iatrogenic trauma during insertion. Instead, simple bends in microwires, recently shown to be effective for the activation of CNS neurons (**Lee et al., 2016**; **Lee and Fried, 2017**; **Lee et al., 2019**; **Ryu et al.,**

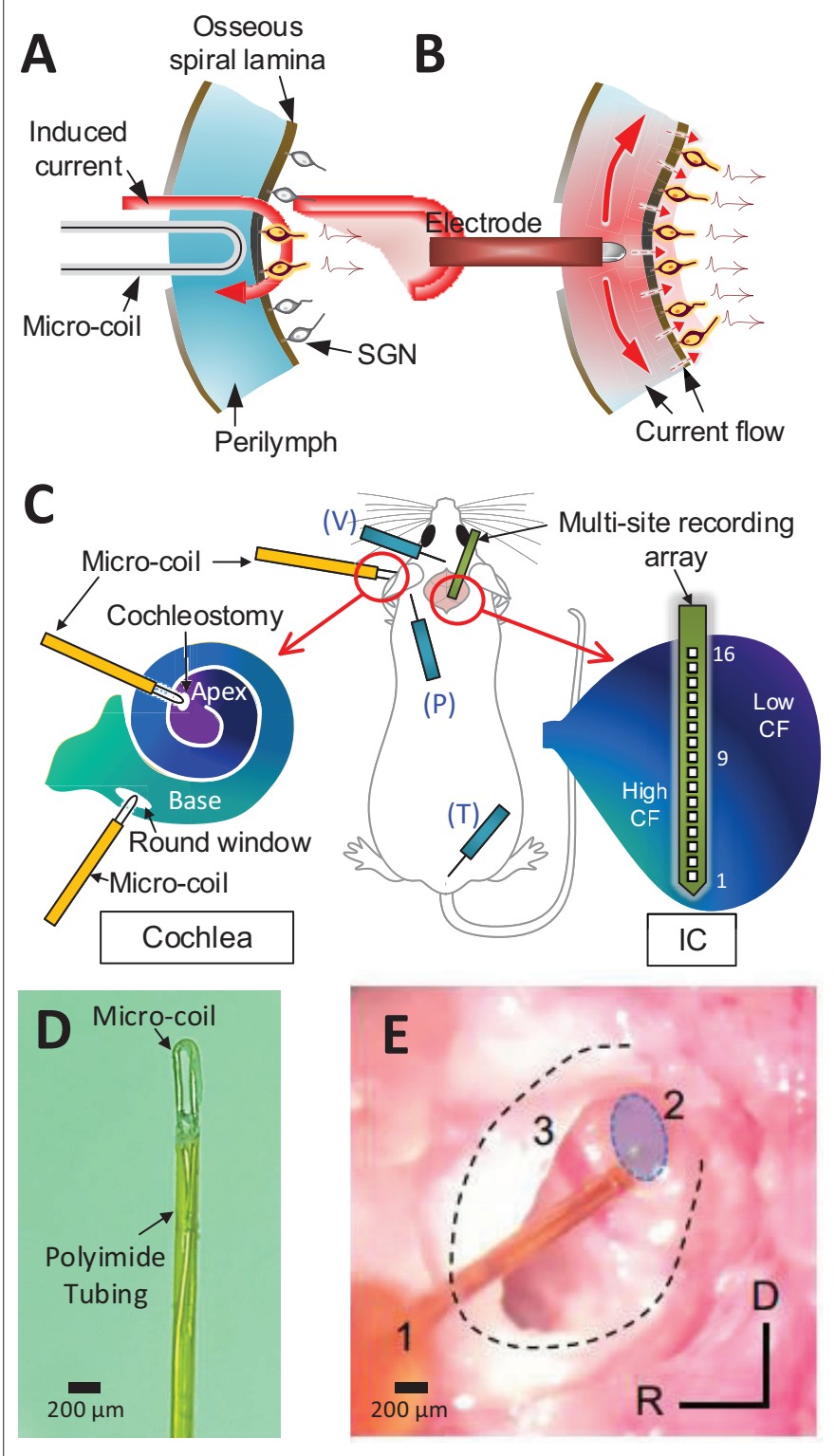

**Figure 1.** Magnetic stimulation of spiral ganglion neurons (SGNs). Conceptual diagram illustrating induced electric field generated by a micro-coil (**A**) and current spread from an electrode (**B**) positioned in the scala tympani. While high conductivity of the perilymph within the scala tympani leads to an expansive spread of current (**B**), the induced electric field is confined to tight regions around the coil (**A**) – see text. (**C**) Schematic of the experimental setup depicting the micro-coil (orange) inserted into the cochlea (basal and apical turns), the multisite recording array (green) inserted across the tonotopic axis of the inferior colliculus, and placement of three subdermal

*Figure 1 continued on next page*

*Figure 1 continued*

recording electrodes (blue) into the vertex (V), pinna (P), and tail (T). (**D**) Photograph of the tip of the micro-coil used in experiments. (**E**) Photograph of the micro-coil (1) inserted through the round window of the left cochlea into the basal turn (2, shaded blue). The stapedial artery (3) is visible. The outline of the cochlea is approximated by dashed lines. Axes: R: rostral, D: dorsal.

*2020*), may offer an attractive alternative to multiturn spiral coil designs because they allow coil sizes to be minimized, and thus the flexibility and overall structure of coil-based CIs can be made to match existing implants, thereby reducing a barrier to implementation.

Here, we investigate the ability of magnetic stimulation from bent-wire micro-coils to drive the auditory pathways. We evaluated the efficacy of stimulation and the resulting spectral spread of activation by recording with a multielectrode array positioned along the tonotopic axis of the inferior colliculus (IC), an auditory nucleus downstream from the cochlea and found in the midbrain. We show that bent-wire micro-coils can indeed drive auditory circuits effectively and further, that the resulting activation from single micro-coils is significantly narrower than that from traditional electrodes, that is, approaching the relatively narrow spread produced by acoustic stimuli. Control experiments verified that responses from coils were indeed magnetic in origin and that they did not arise from activation of hair cells. Taken together, our results suggest that further investigation of coil-based CIs is warranted as they may produce a larger number of independent spectral channels than electrode-based CIs and thus could lead to improved clinical outcomes.

## Results

### Responses to acoustic stimulation in hearing animals

We stimulated the left cochlea with acoustic, electric, and magnetic stimuli and measured responses from a 16-channel recording array implanted along the tonotopic axis of the right (contralateral) inferior colliculus (IC) in anesthetized mice (*Figure 1C*; Materials and methods). Acoustic stimuli consisted of a series of single frequencies ranging from 8 to 48 kHz, chosen to cover much of the tonotopic range represented in the central division of the mouse IC (*Stiebler and Ehret, 1985*). Multiunit activity (MUA) recorded from each of the sites in the IC was quantified by analog representation (*Figure 2—figure supplement 1*), and the cumulative discrimination index, $d'$ ($d$-prime), was calculated to construct spatial tuning curves (STCs; Materials and methods). The MUAs, which are combined responses from single-unit activities with various tuning curves, typically have 'V'-shaped tuning curves (*Snyder et al., 2008*) and MUA-based STCs for acoustic stimulation also have narrow 'V'-shaped curves; examples for 8, 16, and 32 kHz are shown in *Figure 2A–C*, respectively. At $d'$ equal to 1, the channels with the lowest threshold ('best' site, BS) were 12, 9, and 5, respectively (indicated by white stars), which is consistent with the known tonotopic organization of the mouse IC (*Stiebler and Ehret, 1985*). Only 5 out of 11 animals had best sites at 8 kHz (*Figure 2D*) and thus the sampling of the lowest frequency region of the IC may have been incomplete. The data for all animals and frequencies tested also show this tonotopic organization (*Figure 2D*).

Consistent with previous studies, increases in suprathreshold sound pressure levels (SPLs) typically led to increases in the magnitude of the IC response before saturating at higher intensity levels (*Figure 2E*). The dynamic range (DR), defined as the range of stimulus amplitudes for which response strength was between 10% and 90% of the maximum response at BS, averaged 25.96 ± 9.17 dB (Figure 6D), consistent with previous reports in mice (*Köppl and Yates, 1999*; *Nizami, 2002*; *Shivdasani et al., 2008*). To facilitate comparison of DRs across experiments, especially subsequent responses to magnetic and electric stimulation, the stimulus amplitude that elicited 50% of the maximum response was normalized to a level of 0 dB and the plots of response magnitude vs. stimulus level for individual animals were overlaid (*Figure 2E* inset, gray lines). This provides a visual representation of DRs across a given modality; the solid red line is the best-fit sigmoidal curve to all data points, and the dotted lines indicate the 10% and 90% levels, providing a measure of the DR for the visual overlay of the inset.

### Robust activation of the auditory system by magnetic stimulation

After recording auditory brainstem response (ABR) and IC responses to acoustic stimulation (*Figure 3A*), the cochlea was surgically exposed and lesioned to prevent the possibility of acoustic

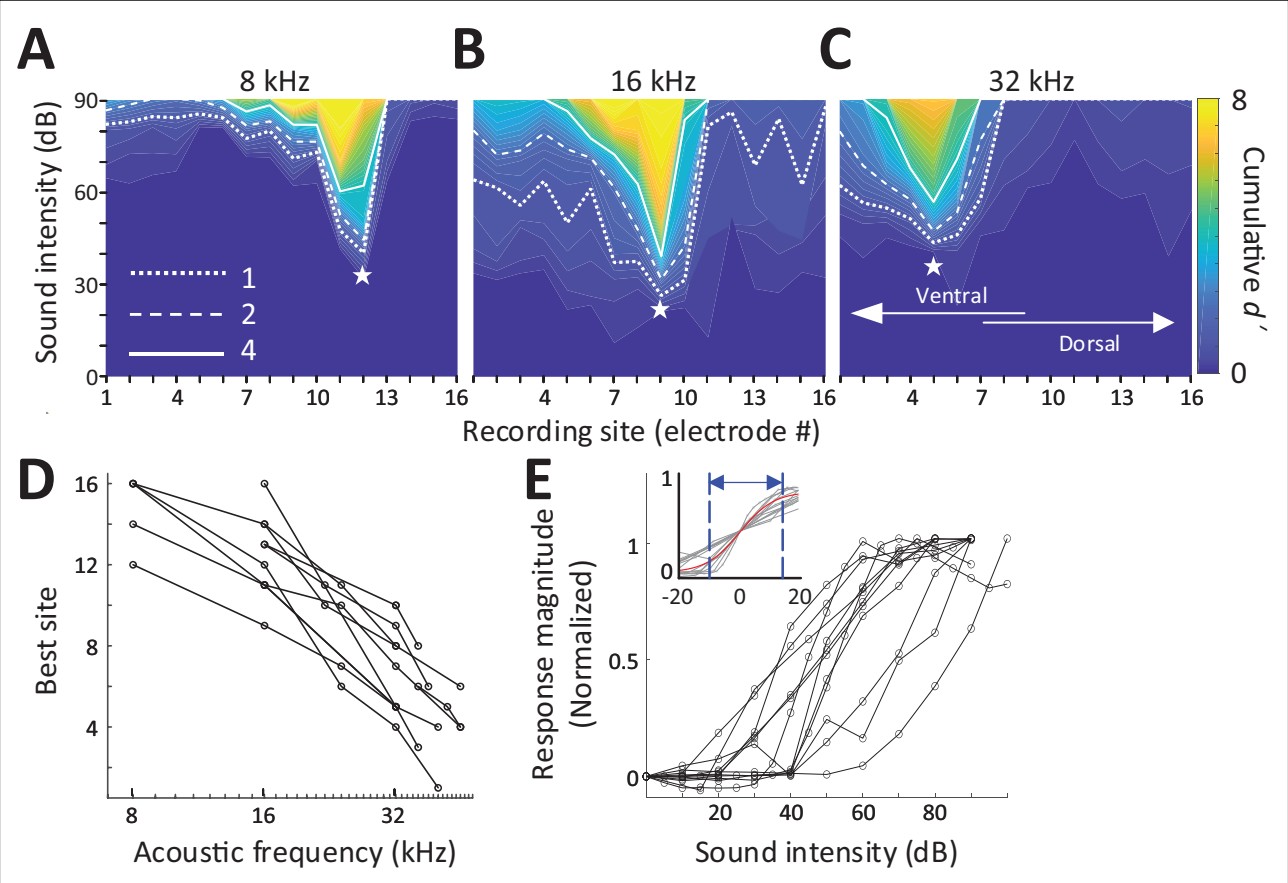

**Figure 2.** Response to acoustic stimulation measured in the inferior colliculus (IC). (**A–C**) Typical spatial tuning curves (STCs) of the IC responses to acoustic stimulation (8, 16, and 32 kHz, respectively) recorded from the 16-channel probe positioned in the IC. Response magnitude was quantified with d-prime analysis (see Materials and methods). The recording site number (x-axis) increases from the IC's ventral to the dorsal end (low to high characteristic frequency). The recording electrode with the lowest threshold (best site, BS) is marked with a white star. Dotted, dashed and solid lines correspond to cumulative d' levels of 1, 2, and 4. (**D**) BS for acoustic stimulation from 8 to 48 kHz; lines connect all data from single animals (n = 11). This mapping is used to assign a 'characteristic frequency' to each electrode. (**E**) Rate-level functions at BS to 32 kHz normalized to peak rate; individual lines are averaged response from individual animals. Inset plots the same data but normalized such that 50% of the amplitude level that elicited the peak response was assigned the level of 0 dB; the solid red line shows the best-fit sigmoidal curve to all data points.

The online version of this article includes the following figure supplement(s) for figure 2:

**Figure supplement 1.** IC responses to acoustic, magnetic, and electric stimulation.

**Figure supplement 2.** Cumulative d' index with respect to dB levels above threshold.

responses arising from magnetic or electric stimulation. For lesioning, distilled water was injected through the round window membrane (see Materials and methods) to induce an osmotic shock to the hair cells (*Chung et al., 2016*). Recordings of ABRs (*Melcher et al., 1996*; *Biacabe et al., 2001*) in mice following intracochlear water instillation demonstrated minimal or no responses up to 75 dB SPL or higher (*Figure 3A, B*).

Following confirmation of severe to profound sensorineural hearing loss on ABR testing, we measured IC responses to both magnetic or electric stimulation, delivered to both basal and apical cochlear. Prior to capturing IC responses, ABRs were recorded each time a coil or electrode was inserted (or reinserted) into the cochlea (see Materials and methods) to provide a relatively quick validation that a given surgical procedure had not damaged the early auditory pathways or the implant itself. ABRs to electric stimulation (eABRs; *Figure 3D*) were generally similar to those reported previously in mice (*Navntoft et al., 2019*) and other laboratory animals (*Miller et al., 1995*), with multiple peaks occurring within the first few milliseconds following stimulus onset. ABR waveforms to magnetic stimulation (mABR; *Figure 3C*) also consisted of a number of peaks, although the amplitudes of the early and later peaks were about the same. The overall appearance of mABRs was closer to that of

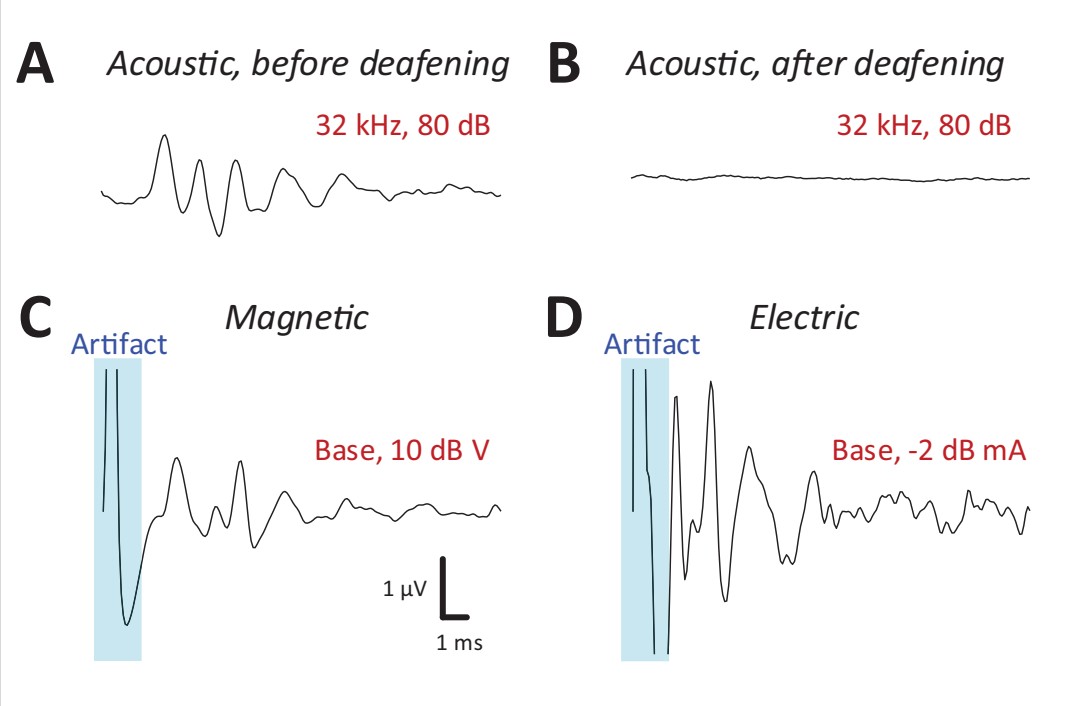

**Figure 3.** Magnetic stimulation evoked robust auditory brainstem responses (ABRs). ABR responses to a 32-kHz monotone: (A) control, (B) after DI water was injected into the cochlear through the round window. ABRs from magnetic (C) and electric (D) stimulation (post-deafening). The blue shaded regions identify the portion of the recording obscured by the stimulus artifact.

The online version of this article includes the following figure supplement(s) for figure 3:

**Figure supplement 1.** Auditory brainstem responses (ABRs) in response to magnetic and electric stimulation.

acoustically evoked ABRs (aABRs) vs. eABRs, although we did not attempt to quantify this observation. Regardless, the generation of robust ABRs to magnetic stimulation strongly suggested that micro-coils can indeed drive the early auditory pathways, and therefore, we proceeded to collect responses from the IC.

Consistent with the presence of robust mABRs, magnetic stimulation also elicited robust neural activity in the IC (*Figure 4A, B*). The typical raw response to each modality is shown in *Figure 2—figure supplement 1*. Responses to magnetic stimulation were consistent with the tonotopic organization of the cochlea. Responses to stimulation of the basal turn were strong in the ventral portion of the IC, the region known to process high frequencies, with little or no responses observed outside this region (*Figure 4A*). Across animals (*n* = 6), the average characteristic frequency for BSs (extrapolated from *Figure 2D*) in response to micro-coil-based stimulation of the basal turn was 37.36 ± 4.00 kHz. In contrast, magnetic stimulation of the apical turn elicited only a narrow portion in the dorsal portion of the IC (*Figure 4B*), known to process lower sound frequencies. Averaging across the population, the characteristic frequency of the BSs for magnetic stimulation of the apical turn was 8.44 ± 6.58 kHz.

In contrast to the relatively narrow spectral spread of IC activity arising from magnetic stimulation, the spread from electric stimulation was considerably wider (*Figure 4C, D*), consistent with findings in previous studies (*Shannon, 1983*; *O'Leary et al., 1985*, *O'Leary et al., 2009*, *Micco and Richter, 2006*; *Middlebrooks and Snyder, 2007*). Nevertheless, BSs again showed evidence of tonotopic organization, that is, stimulation of the basal turn was centered in the ventral portion of the IC while stimulation of the apex resulted in activation centered in the dorsal portion.

## Spatially confined responses elicited by magnetic stimulation

To quantify the spread of excitation across modalities, we measured the width of the $d'$ = 1 trace (i.e., the distance between the ventral- and dorsal-most electrodes exhibiting a suprathreshold response) at the stimulus amplitude for which the BS reached $d'$ levels of 2 and 4 (the red arrow in *Figure 4B* illustrates a sample calculation for a $d'$ level of 2) (see Materials and methods). The spatial spread, that

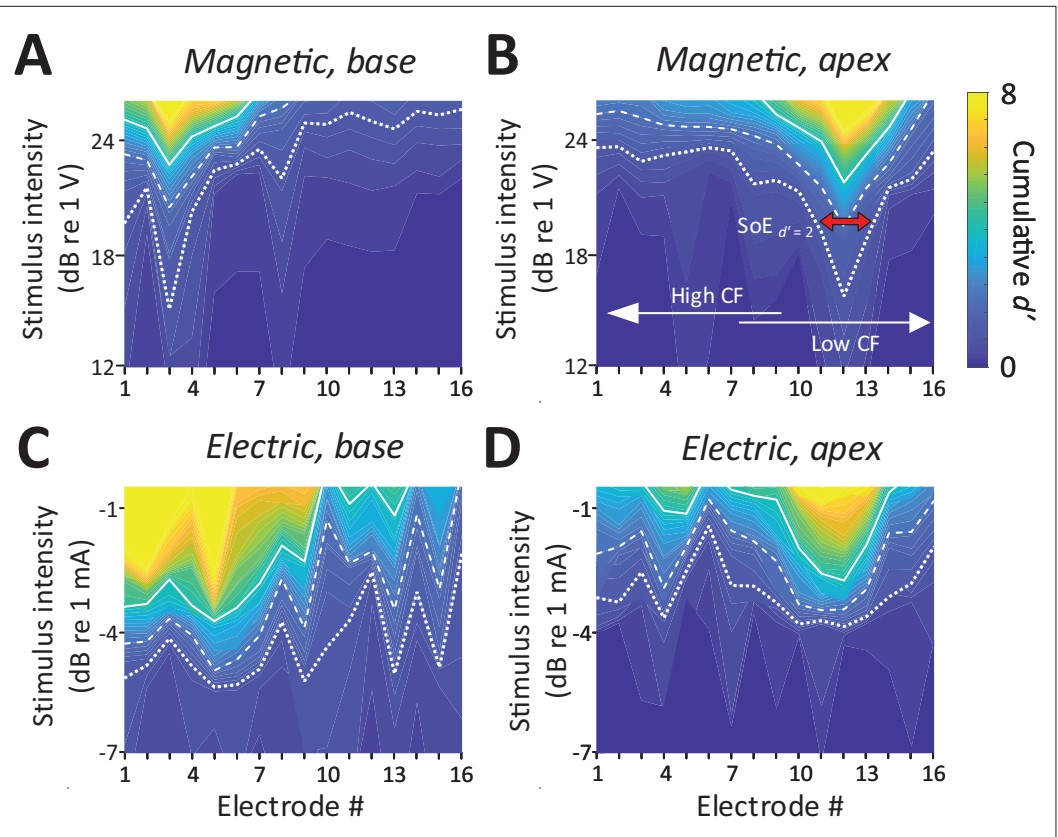

**Figure 4.** Spatial tuning curves (STCs) for magnetic stimulation are spatially more confined than STCs for electric stimulation. (**A, C**) STCs in response to magnetic and electric stimulation delivered to the basal turn of the cochlea (aMUA signals quantified with *d*-prime analysis – see text). Dotted, dashed, and solid lines are contours for cumulative *d'* values of 1, 2, and 4. (**B, D**) STCs in response to magnetic and electric stimulation of the apical turn. The color bar on the right side of panel B applies to all panels. The red arrow in panel B indicates the spread of excitation (SoE – see text).

The online version of this article includes the following figure supplement(s) for figure 4:

**Figure supplement 1.** Spatial tuning curves (STCs) in response to magnetic and electric stimulation at the basal turn of the cochlea.

**Figure supplement 2.** Spatial tuning curves (STCs) in response to apical magnetic and electric stimulation.

is, the distance between the dorsal- and ventral-most responding electrodes (*Figure 5A, B*), was then converted into a spectral spread, that is, the width of the corresponding frequency bands, derived from *Figure 2D*; *Figure 5C, D*.

In all cases, that is, for both basal and apical stimulation and both *d'* levels (moderate and high discrimination levels), the spread of activation from electric stimulation was wider than that from magnetic stimulation (*Figure 5*, *Table 1*). This difference was significant for all stimulation location, except for apical stimulation at *d'* = 2 (p = 0.8287; *Figure 5B*). When the spread of magnetic stimulation was compared to that from auditory stimuli, there were no statistical differences for stimulation of the basal and apical turn (both moderate and high discrimination levels) (p > 0.05, *Table 1*).

The number of peak 'tips' observed in the STCs differed across the stimulus modalities (*Figure 5E*). For example, eight of the nine STC profiles generated in response to electric stimulation exhibited two or more tips leading to an average of 2.33 per profile. In contrast, seven of the nine STC profiles for magnetic stimulation had only a single peak tip, with the remaining two profiles showing double tips (average of 1.22). All profiles for auditory stimulation had a single peak only. The number of tips for electric stimulation was statistically higher than that from magnetic stimulation (p = 0.007) or from acoustic stimulation (p < 0.001). The difference between the number of tips for magnetic vs. acoustic stimulation was not statistically significant (p > 0.05). Taken together, these results indicate that the

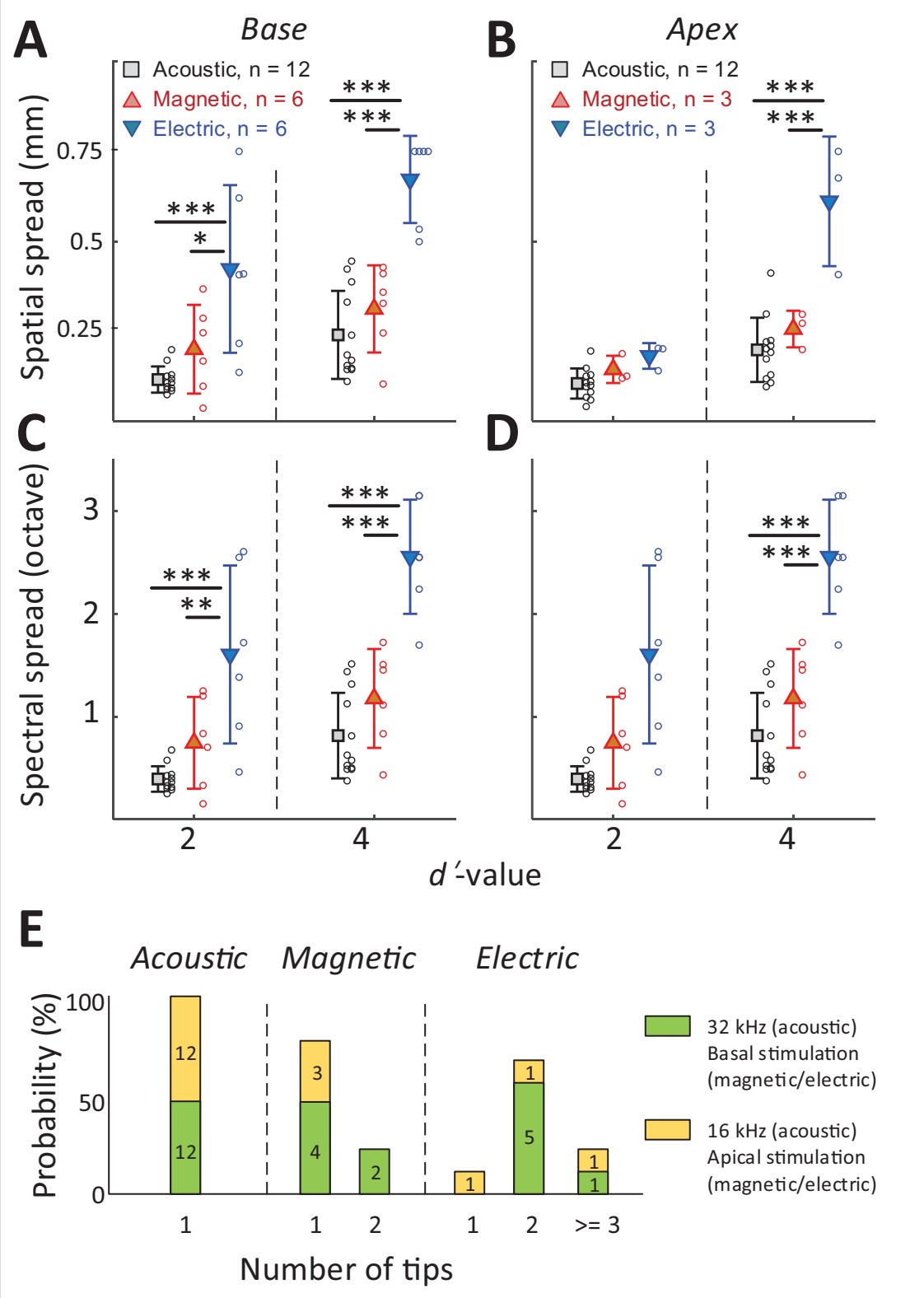

**Figure 5.** The spread of activation is narrower for magnetic vs. electric stimulation. (**A, B**) The spatial spread of IC activation was measured by the width of activated channels at cumulative $d'$ levels of 2 and 4. (**C, D**) The spectral spread was estimated by converting the width of activated channels to octave distance based on the characteristic frequency of each electrode (derived from **Figure 2D**). Two-way ANOVA with subsequent Tukey's test was applied to verify the statistical significance: \*$p < 0.05$, \*\*$p < 0.01$, \*\*\*$p < 0.001$. Error bars indicate mean ± standard deviation. (**E**) The number of tips in the spatial tuning curves (STCs) for each stimulus modality (see text). The numbers inside the bar plots indicate the number of STCs for each group.

**Table 1.** Statistical analysis of spatial spread (*Figure 5A, B*) by two-way ANOVA and subsequent Tukey's test for multiple comparisons; *p < 0.05, ***p < 0.001.

| | | | Predicted (LS) mean diff. | 95.00% CI of diff. | Summary | Adjusted p value |
|---|---|---|---|---|---|---|
| | | Acoustic vs. magnetic | −0.08433 | −0.2383 to 0.06961 | ns | 0.3863 |
| | | Acoustic vs. electric | −0.3076 | −0.4615 to −0.1536 | *** | <0.0001 |
| | *d' = 2* | Magnetic vs. electric | −0.2233 | −0.4010 to −0.04549 | * | 0.0108 |
| | | Acoustic vs. magnetic | −0.07223 | −0.2262 to 0.08171 | ns | 0.4953 |
| | | Acoustic vs. electric | −0.4324 | −0.5863 to −0.2785 | *** | <0.0001 |
| Base | *d' = 4* | Magnetic vs. electric | −0.3602 | −0.5379 to −0.1824 | *** | <0.0001 |
| | | Acoustic vs. magnetic | −0.03936 | −0.1635 to 0.08482 | ns | 0.7172 |
| | | Acoustic vs. electric | −0.07669 | −0.2009 to 0.04749 | ns | 0.2949 |
| | *d' = 2* | Magnetic vs. electric | −0.03733 | −0.1944 to 0.1197 | ns | 0.8287 |
| | | Acoustic vs. magnetic | −0.05788 | −0.1821 to 0.06630 | ns | 0.4922 |
| | | Acoustic vs. electric | −0.4130 | −0.5372 to −0.2889 | *** | <0.0001 |
| Apex | *d' = 4* | Magnetic vs. electric | −0.3552 | −0.5122 to −0.1981 | *** | <0.0001 |

spread of activation was significantly narrower for magnetic vs. electric stimulation, regardless of the location at which stimulation was delivered. While much additional testing is required, the narrower spectral spread from magnetic stimulation suggests the possibility that a coil-based CI may create narrower and more independent spectral channels, thus offering the potential for improved performance of an implant.

## Larger DR with magnetic stimulation

The rate-level functions to magnetic and electric stimulation (measured at the BS) are shown in *Figure 6A, B*, respectively. The average DR across the population was smaller for magnetic stimulation (10.05 ± 4.18 dB V, *n* = 6; basal stimulation, measured at the BS) than for acoustic stimulation (25.96 ± 9.17 dB SPL; 32 kHz tone; p < 0.001) but larger than that for electric stimulation (3.24 ± 0.99 dB mA; p = 0.0031) (*Figure 6D*), suggesting better discrimination resolution for stimulus intensity. The differences in DR did not arise from different neuronal response levels as the maximum response evoked by magnetic stimulation was comparable to that evoked by acoustic or electric stimulation, that is, no statistically significant differences between any pair of modalities (*Figure 6C*). Note that in some experiments with magnetic stimulation, response rates did not saturate, even at the maximum stimulation levels tested here (*Figure 6A*), suggesting that the DRs reported here may be underestimated.

## Responses to micromagnetic stimulation in hair cell ablated animals

As a final control experiment, we confirmed that magnetic and electric responses were similar in animals with chronically lesioned hair cells. We applied a Gelfoam pledget soaked with gentamicin to the round window to cause a complete loss of outer hair cells and near complete loss of inner hair cells in the basal turn of the cochlea (*Figure 7A, B*; *Heydt et al., 2004*). After 10 days, hearing was evaluated with acoustic ABR and distortion product otoacoustic emissions (DPOAEs). DPOAEs were absent and ABR thresholds were beyond the range of the acoustic system or markedly increased (>75 dB SPL); a small amounts of residual ABR can arise from acoustic crosstalk from the contralateral unlesioned ear (*Harrison et al., 2013*). In these gentamicin-treated animals (*n* = 2), responses to electric and magnetic stimulation in the basal turn were robust and spatial spread remained narrower in magnetic vs. electric stimulation (*Figure 7C*). Immunostaining in the basal turn at 32 kHz for hair cells (Myo7a, white) and actin-containing supporting structures (Phalloidin, red) showed no remaining inner and outer hair cells (*Figure 7A*), eliminating the possibility that observed responses to magnetic or electric stimulation arose from inadvertent activation of hair cells.

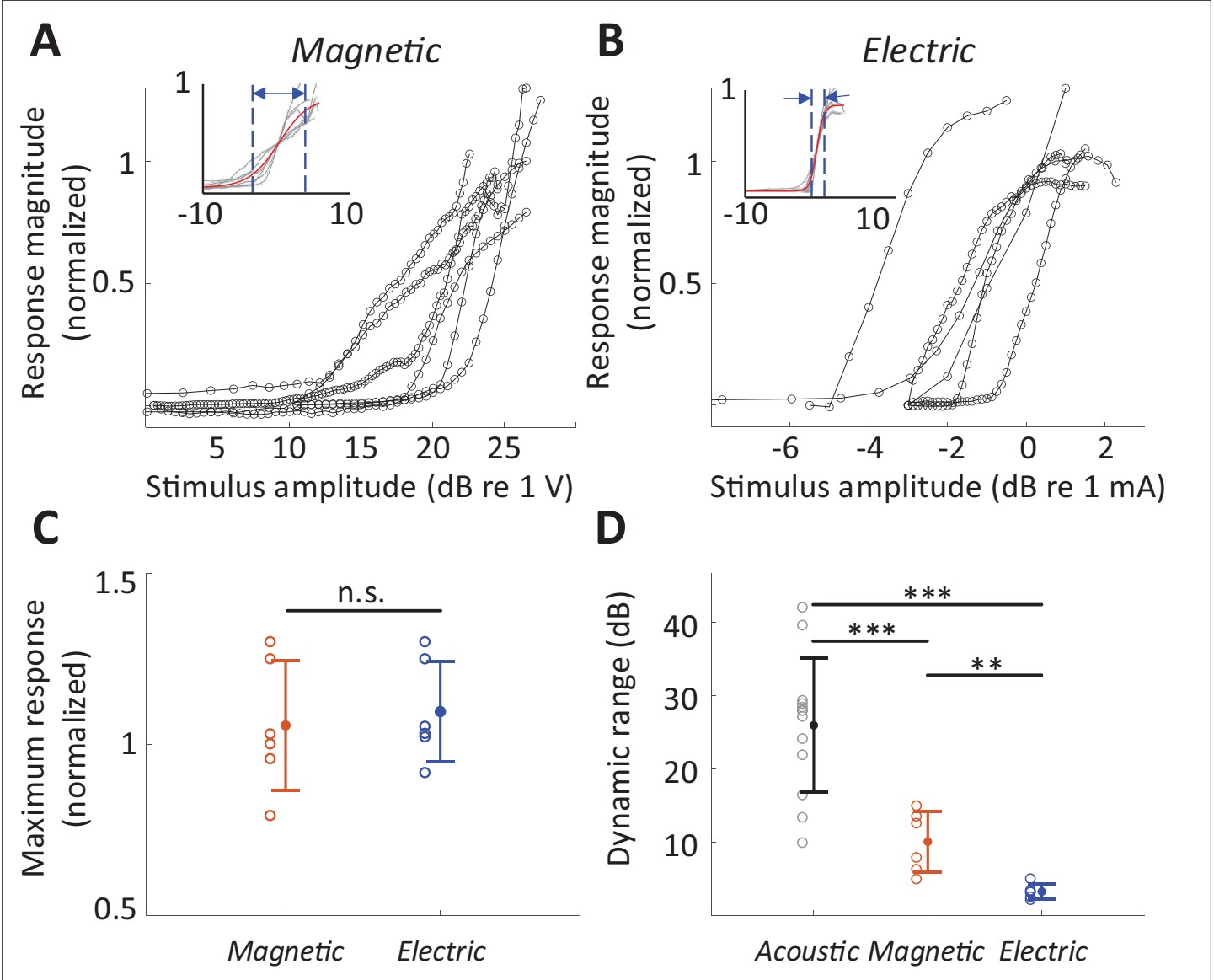

**Figure 6.** Dynamic range for magnetic stimulation is wider than that for electric stimulation. Normalized responses rates as a function of stimulus intensity for magnetic (**A**) and electric (**B**) stimulation (basal turn). Each line is the averaged response curve from one animal. Insets show the same data normalized such that 50% of the peak response was assigned the level of 0 dB; the red line shows the best-fit curve to all raw data points. (**C**) Individual points are the distribution of the maximum response rates to magnetic and electric stimulation. Error bars indicate mean ± standard deviation. Each response rate was normalized by the maximum response to acoustic stimulation obtained from the same animal. (**D**) Individual points are the distribution of dynamic ranges for each mode of stimulation. Error bars indicate mean ± standard deviation. A Student $t$-test was applied to verify the statistical significance: **$p < 0.01$, ***$p < 0.001$.

## Discussion

We used a combination of ABR measurements and multiunit recordings to demonstrate that magnetic stimulation, delivered from a bent-wire micro-coil inserted into the cochlea, can effectively drive the auditory pathways. Magnetic stimulation evoked multipeaked ABRs, suggesting that coil-based stimulation was indeed capable of activating SGN processes that, in turn, led to activation of central nuclei in the auditory pathway. Multielectrode recordings obtained in the IC showed that responses were robust, narrowly confined, and tonotopically organized. The responses from basal and apical micro-coil locations were narrow and showed little or no overlap between them, whereas responses from electric stimulation showed considerable overlap for stimulation at the same locations. The number of peak tips observed in the ST profiles generated from the IC recordings was also significantly lower

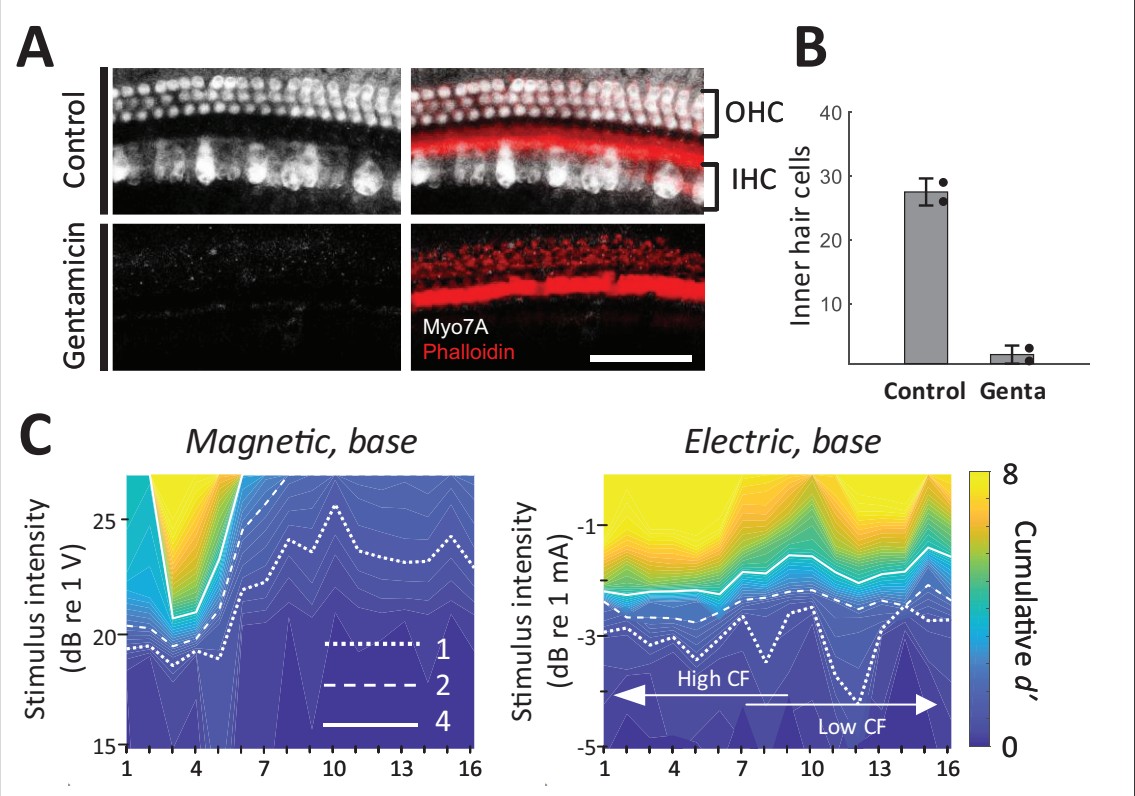

**Figure 7.** Magnetic stimulation elicits responses in chronically deafened mice. (**A**) Gentamicin caused complete loss of inner and outer hair cells (white, Myo7A). Supporting structures of the organ of Corti were stained with Phalloidin (red). Scale bar: 50 μm. (**B**) Quantification of inner hair cells in the basal turn. Hair cells per 100 μm were counted at 32, 45.25, and 64 kHz. Error bars indicate mean ± standard deviation (*n* = 2 animals). (**C**) Spatial tuning curves (STCs) of the IC responses to magnetic and electric stimulation at the base, measured in a gentamicin-treated mouse. Dotted, dashed, and solid lines are contours for cumulative *d′* values of 1, 2, and 4, respectively. Data availability: the source data and codes are available on the Open Science Framework (DOI 10.17605/OSF.IO/Y7ZRX).

for magnetic stimulation than for electric stimulation. Suggesting the spread to encompass additional cochlear turns was less with magnetic stimulation. Taken together, the results strongly suggest that coil-based activation of the cochlea is more narrowly confined than that of electric stimulation.

A number of control experiments and experimental safeguards were used to verify that observed responses arose from magnetic stimulation of SGNs and not from other factors. First, the impedance from the coil to ground was monitored before and after each experiment and remained consistently above 200 MΩ, eliminating the possibility that observed responses arose from direct electrical activation, for example, from leakage of the stimulus current into the surrounding perilymph. If the impedance of the coil insulation was found to be than 200 MΩ after an experiment, recordings obtained from the coil were exempted from the analysis. Second, the direct current (DC resistance across the coil leads was also monitored regularly and it too remained stable (<10 Ω)), eliminating the possibility that a broken coil might be activating neurons capacitively. The coils with DC resistance over 10 Ω were discarded. Third, measurements of the temperature change produced by coils were previously shown to be less than 1°C (*Lee et al., 2016*; *Ryu et al., 2020*), greatly reducing the possibility that observed responses arose from some type of temperature shock. The fourth set of control experiments arose from concerns that observed responses could be mediated through activation of hair cells, for example, micromovements of the coil during the delivery of stimulus current could result in transmission of a pressure wave through the scala tympani (*Kallweit et al., 2016*). To eliminate this possibility, we injected DI water into the cochlea after completing the measurements of auditory responses. The resulting osmotic shock led to loss of responses to subsequent auditory stimuli, even at SPL levels of 75 dB, strongly suggesting a loss of hair cell functionality. To provide even stronger assurance, we injected gentamicin into the cochlea in a subset of animals; robust responses to magnetic and electrical stimulation persisted in these animals but not to acoustic stimulation. Postmortem

immunochemical staining of the cochlea revealed a complete absence of hair cells, eliminating the possibility that responses arising from micro-coils were mediated through a magnetoacoustic effect.

## Activation from magnetic stimulation is spatially confined

To compare the spread of activation across modalities, we quantified MUAs recorded at multiple sites along the tonotopic axis of IC and constructed STCs (Materials and methods). The MUAs, which are combined responses from single-unit activities with various tuning curves, typically have 'V'-shaped tuning curves (*Snyder et al., 2008*) and MUA-based STCs for acoustic stimulation also have narrow 'V'-shaped curves (*Snyder et al., 2008*).

Previous studies have reported that the spread of current from electrodes inserted into the scala tympani recruits SGNs across spatially extensive regions, thereby limiting the spectral specificity of artificial sound encoding (*Shannon, 1983*; *O'Leary et al., 1985*, *O'Leary et al., 2009*, *Micco and Richter, 2006*; *Middlebrooks and Snyder, 2007*). For example, a study in cat (*Middlebrooks and Snyder, 2007*) compared STCs for acoustic stimulation to those for monopolar and bipolar electric stimulation. Although the STCs for bipolar electric stimulation were significantly narrower than those for monopolar stimulation in that study, both were substantially wider than STCs for acoustic stimulation (8.23- and 4.25-fold, respectively). Another study in mice (*Thompson et al., 2020*) reported that STCs for electric stimulation were three to four times broader than those for pure tones. Thus, the wide spread of activation from electric stimulation measured here (3.14 times larger than that from monotones) is in good agreement with previous reports. Computational models that explore the reasons for the lack of confinement with electric stimulation find that the high conductivity of the perilymph in the scala tympani causes significant spread of the electric field arising from each electrode (*Vanpoucke et al., 2004*; *Briaire and Frijns, 2000*; *Frijns et al., 2009*; *Jolly et al., 1996*). In addition, because the SGN processes targeted by stimulation are situated on the other side of the high-resistance bony wall of the scala tympani, it is necessary to employ stronger stimulus amplitudes, thereby exacerbating the spread.

In contrast to the electric fields arising from electrodes, the spread of activation from coils measured here was significantly narrower. While our experiments do not identify the reasons for the lower spread, magnetic stimulation is known to have some intrinsic advantages over electrodes and electric stimulation. For example, the physics underlying the generation and spread of magnetic fields (Maxwell's equations) ensures that the induced electric fields are confined to tight regions around the magnetic flux. In addition, the magnetic fields induced by the flow of electric current through coils are highly permeable to all biological materials and thus are relatively impervious to bone and other high-resistance materials within the cochlear environment. As a result, magnetic fields pass readily through the high-resistance bony wall of the scala tympani, without the need to increase stimulation amplitudes (thus, limiting additional spread). Although magnetic fields are not thought to activate neurons directly, time-varying magnetic fields induce electric fields, and therefore magnetic fields 'carry' the electric field across the walls of the scala tympani where they can induce activation of SGN processes.

## Improved DR with magnetic stimulation over electric stimulation

In addition to the low spectral resolution associated with electric stimulation, the DR for encoding sound intensity is also limited. For example, the DR for listeners with normal hearing is approximately 120 dB while the DR for CI users is typically restricted to 10–20 dB (*Hong et al., 2003*). The small DR for electric stimulation is related in part to the wide spread of activation (*Viemeister, 1988*). SGN populations with similar characteristic frequencies encode sound intensity together which allows a wider DR to be perceived at downstream auditory circuits (*Sachs and Abbas, 1974*; *Viemeister, 1988*; *Hudspeth, 2014*); simultaneous activation of all such fibers with electric stimulation compresses the DR. The relatively small DR for electric stimulation results in the need for amplitude compression of the acoustic signal with CIs (*Hong et al., 2003*). The practical implications of this compression are potential decreases in speech recognition (*Fu and Shannon, 2000*; *Loizou et al., 2000*), particularly in the presence of increased background noise (*Zeng and Galvin, 1999*), as well as reductions in sound quality (*Boike and Souza, 2000*; *Neuman et al., 1998*).

Our measurements suggest that the DR for magnetic stimulation from micro-coils was approximately three times greater than that for electrical stimulation. It is likely that the gradual recruitment of SGNs with magnetic stimulation contributed to the wider DR. Further, the responses to magnetic

stimulation were not saturated at the peak stimulus amplitudes we used here, suggesting the DR values reported here may be underestimated. The expanded DRs for magnetic stimulation provide additional support for the potential of micro-coil-based CIs to enhance the quality of CI-induced hearing.

## Future efforts and limitations of micro-coil stimulation of the cochlea

The ability to create narrow spectral channels with micro-coils, even in the tiny cochlea of the mouse, is encouraging as it suggests that coil-based stimulation has the potential to create a larger number of independent spectral channels in clinical use. Further, the relatively simple and compact bent-wire coils used here raise the possibility that CIs can be manufactured that are structurally similar to existing devices. This too is encouraging because such an approach raises fewer short- and long-term safety concerns. While much additional development and testing is needed, we believe that these findings clearly suggest that further investigation of coil-based stimulation of the cochlea is warranted.

In this study, electric stimulation was delivered in a monopolar configuration. Other configurations, for example, bipolar, tripolar, and focused multipolar result in improved spatial selectivity in both animal models (*Snyder et al., 2008*; *Bierer et al., 2010*; *George et al., 2015*) and human trials (*van den Honert and Kelsall, 2007*) although at the expense of increased thresholds (*Bierer and Faulkner, 2010*; *Zhu et al., 2012*; *George et al., 2015*). Due to the small size of the mouse cochlea, it was not feasible to test configurations that required the insertion of two or more electrodes into the cochlea. In addition, the advantage of multipolar stimulation is less obvious in species with smaller cochleae, for example, even in the gerbil cochlea difference in spatial spread between monopolar and bipolar stimulation was not significant (*Dieter et al., 2019*). Nevertheless, it will be still interesting to compare spreads from micro-coils to the diverse configurations of electric stimulation in future studies. It is also important to note that the small size of the mouse cochlea limited our choice for the stimulating electrode, while we used a small conical electrode (a height of ~125 µm and a base diameter of 30 µm), commercially available CIs consist of much larger planar electrode (~0.5 mm × 0.5 mm). Due to the high impedance associated with the small electrode size and difference in electrode geometry, it is likely that the spread of electric stimulation presented in this study underestimated those from conventional CIs. Thus, we are planning to compare the performance of micro-coils to clinical devices in the future study with larger animals.

Despite the encouraging results in this first assessment of coil-based stimulation of the cochlea, several key elements of micro-coil design and performance will need to be optimized before they can be considered for human trials. For example, the amplitude of the electric current that flowed through the coil was typically quite large, ~770 mA at threshold, raising concerns about power consumption and battery life. While the relatively low impedance of micro-coils (typically ~10 Ω) helps to reduce the $I^2 \times R$ power consumption of coil-based devices and thus compensates somewhat for the high current levels, supplied electric energy for a single magnetic 'pulse' (52 µJ) is still considerably higher than that for a single electrical biphasic pulse (245 nJ – based on 700 µA and 10 kΩ impedance). It is likely that power consumption in future micro-coil devices can be significantly improved using a number of changes that are relatively straightforward to implement. For example, the coil design used here was identical to that used for stimulation of cortex; tailoring the coil design to optimize SGN activation could potentially reduce power consumption by an order of magnitude or more (*Lee et al., 2019*; *Frijns et al., 2001*). In addition, switching from the platinum–iridium wires used here to higher conductivity materials such as silver or gold could reduce power consumption in half. Finally, the incorporation of magnetic cores into the coil could potentially reduce thresholds by several orders of magnitude. Importantly, future testing will need to take place in animal models whose cochleae better resemble those of humans. Larger size scala tympani associated with such animals will likely require stronger activation thresholds to compensate for the increased distance to targeted neurons, although they will also allow larger wire sizes, possibly offsetting the difference. The high stimulus amplitudes associated with coils also raise concerns about electrical safety, although it is important to remember that the flow of electrical current through the coil is electrically isolated from the surrounding tissue. Advanced control circuits can be incorporated into future designs to further minimize the potential for tissue damage. Even without the electrical concerns, the high current levels required to activate SGNs may produce temperature changes that could exceed safe limits; these too will need to be evaluated prior to chronic use of coil-based implants.

Another potential concern would be the compatibility of implanted micro-coils with strong exogenous magnetic fields. A previous study has tested the effect of 1.5 T exogenous magnetic field on micro-coils and electric wire-based implants designed for deep brainstem stimulation (**Bonmassar and Serano, 2020**). Their results showed warming of the implants in both groups, however, the degree was far less in the micro-coils (<1°C) than in the electric wires (10°C). Nevertheless, testing the effect of exogenous magnetic fields on coil-based CIs will be crucial for the translation of this technique to humans.

In the present study, it is likely that most SGNs were intact since our deafening procedure mainly targeted hair cells. Maintaining uniform survival of SGNs was essential to ensure accurate comparison of the spread of activation across modalities, however, this situation does not uniformly reflect the pathological conditions of all implanted patients. Patients typically receive CIs months or years after the onset of deafness and often have considerable SGN loss (**Nadol and Eddington, 2006**; **Khan et al., 2005**). Thus, in future studies, it will be necessary to test coil effectiveness in neonatally deafened animals so as to more closely mimic pathological conditions of implanted patients.

Finally, it will also be necessary to build and test devices that incorporate multiple coils to ensure that they can indeed be reliably developed and safely implanted as well as to determine the minimum separation for which coils remain independent. The ability to match the structure and mechanical properties of coil-based CIs will be highly beneficial as it will allow existing fabrication techniques and surgical procedures to be harnessed, thereby facilitating the transition into clinical practice.

# Materials and methods

## Key resources table

| Reagent type (species) or resource | Designation | Source or reference | Identifiers | Additional information |
|---|---|---|---|---|
| Strain, strain background (*Mus musculus*) | CBA/CaJ | The Jackson Laboratory, Bar Harbor, ME | 000654 | |
| Antibody | Anti-myosin 7A (rabbit polyclonal) | Proteus Biosciences, Ramona, CA | 25-6790 | 1:200 |
| Antibody | Anti-Rabbit IgG (H + L) Cross-Adsorbed Secondary Antibody, Alexa Fluor 488 (goat polyclonal) | Invitrogen, Carlsbad, CA | A-11008 | 1:500 |
| Chemical compound, drug | Gentamicin Sulfate | Sigma-Aldrich, St. Louis, MO | G-4918 | 200 µg |
| Software, algorithm | MATLAB | MathWorks | RRID:SCR_001622 | |
| Software, algorithm | GraphPad Prism | GraphPad | RRID:SCR_ 002798 | |
| Other | Alexa Fluor 647 Phalloidin | Invitrogen, Carlsbad, CA | A22287 | 1:200 |

## Animal preparation

All procedures were approved by the Institutional Animal Care and Use Committee of Massachusetts Eye and Ear (protocol# 15-003), and carried out in accordance with the NIH Guide for the Care and Use of Laboratory Animals. CBA/CaJ mice were purchased from Jackson Laboratories or bred in house. Mice of either sex aged 6–16 weeks were used in the experiments. Experiments were conducted in an acoustically and electrically isolated walk-in chamber kept at 32–36°C. Mice were anesthetized for the duration of experiments with ketamine (100 mg/kg) and xylazine (10 mg/kg). Animals' anesthesia level and heart rate were regularly monitored, and one-third of the initial ketamine/xylazine dose (i.e., 33 and 3 mg/kg, respectively) was given as needed.

The left cochlea was accessed surgically. A postauricular incision was made and the underlying tissue and musculature were dissected to expose the bulla. A bullotomy was performed by carefully rotating a 28 G needle and enlarging the hole with fine forceps to expose the round window. The left pinna with skin and tissue extending into the external auditory canal was cut and removed to expose the tympanic membrane. To access the contralateral inferior colliculus (IC), the postauricular incision was extended to over midline, and a craniotomy was made just caudal to the temporoparietal suture and to the contralateral side of the midline with a scalpel.

After the original insertion into the inferior colliculus, the position of the multielectrode recording array was not repositioned while switching from one stimulus modality to the next (acoustic, electric, magnetic). Due to the fragility of the recording electrode array, we took extra care to avoid disturbing the skull, as dislodgement of the array would have altered tonotopicity and thus weakened the ability to accurately compare spectral spread between trials.

## Stimulation

All stimuli were generated using LabVIEW and MATLAB software controlling custom-made system based on National Instruments 24-bit digital input/output boards.

### Acoustic stimulation

A custom acoustic system coupled to a probe tube was inserted into the external ear canal close to the tympanic membrane, with two miniature earphones (CDMG150 008-03A, CUI) serving as sound sources. Acoustic stimuli were pure tone pips of 5-ms duration.

### Magnetic and electric stimulation

Magnetic stimulation was presented using custom-made micro-coils (MicroProbes, Gaithersburg, MD), which are highly similar to ones used in previous studies for stimulation of other, noncochlear regions of the CNS (*Lee et al., 2016*; *Ryu et al., 2020*). The coil was fabricated by bending a 25-µm-diameter platinum–iridium into a U-shape (*Figure 1D*). The length of the coil was 3 mm and the width was 175 µm. The DC resistance of the coil was in the range of 8–10 Ω. The coil wire was coated with 5-µm-thick parylene for electrical insulation. Coils were tested before and after each experiment to ensure that there was no inadvertent leak of electric current from the coil to the cochlea. To this end, each coil was submerged in NaCl solution (0.9%) and the electric resistance between one of the coil's terminal ends and an electrode immersed in the solution was measured; resistances above 200 MΩ were considered sufficient for insulation. At least three individual coils with an identical design were tested. To deliver magnetic stimulation, a micro-coil was inserted through the round window, and responses to a range of stimulation parameters delivered to the basal turn were captured. After completion of experiments at the basal turn of the cochlea, a cochleostomy was performed near the apex, allowing analogous experiments to be performed at the apical turn as well. The stimulus was generated by a function generator based on National Instruments 24-bit digital input/output boards and amplified by a voltage amplifier with a gain of 9 V/V and a bandwidth of 70 kHz (PB717X, Pyramid Inc, Brooklyn, NY, USA). The voltage amplifier was powered by a commercial battery (LC-R1233P, Panasonic Corp., Newark, NJ, USA). The stimulus waveform was a positive-going ramp with a rise time of 25 µs. The fall time was set to 0 µs but was limited by the sampling rate of the hardware (100 kHz). The amplitude of the waveform from the function generator was 0 to 1.7 V. The output of the amplifier was 0–15.3 V. The peak levels of magnetic stimulation are limited by Joule heating of the small wires that comprise the micro-coils and the resulting potential to induced failure. At a given stimulus strength, the waveform was presented a minimum of 39 times with a pulse rate of 25 pulses/s.

Once completed, the coil was removed and replaced with a micro-electrode so that an analogous series of electric stimulation experiments could be performed at the same location. Electric stimulation was delivered in a monopolar configuration. The stimulating electrode was a conical platinum–iridium tip with a resistance of 10 kΩ, a height of ~125 µm and a base diameter of 30 µm (Microprobes for Life Science, Gaithersburg, MD, USA; PI2PT30.01 A10). An EMG needle was inserted into the neck muscle to serve as a return electrode. At least three individual electrodes with an identical design were tested. Like micro-coils, the stimulating electrodes were inserted through the round window (for basal stimulation) or via cochleostomy in the apical turn for intracochlear stimulation. Electric stimuli were also controlled by the same function generator used for magnetic stimulation. Stimulus waveforms were rectangular biphasic pulses with phase duration of 25 µs and no interphase interval. Stimulation amplitudes ranged from 0 to 1000 µA. The stimulus for each amplitude was repeated at least 39 times with a pulse rate of 25 pulses/s.

Magnetic stimulation responses were assayed first in some animals, whereas electric stimulation responses were assayed first in other animals. The same placements were used for either the micro-coils or the electrodes. Experiments were terminated whenever the animal's vital parameters, as measured by heart and respiratory rate, declined. The decline was typically observed at around

5–7 hr and preceded by a decline in inferior colliculus responses. A complete set of basal and apical stimulations were tested in two animals, a complete set of only basal stimulation were tested in four additional animals, and a complete set of only apical stimulation were tested in one additional animal.

## Data acquisition and analysis

### DPOAE and ABR

DPOAE and ABR were recorded as previously described (*Seist et al., 2020*). A custom acoustic system was inserted into the external ear canal close to the tympanic membrane. DPOAEs were measured as ear canal pressure in response to two tones presented into the ear canal ($f_1$ and $f_2$, with $f_2/f_1$ = 1.2 and $f_1$ being 10 dB above $f_2$) at half-octave steps from $f_2$ = 5.66–45.25 kHz, and in 5 dB intensity increments from 10 to 80 dB SPL. DPOAE thresholds were defined as the $f_2$ intensity required to generate a DP response of 10 dB SPL over noise floor. ABR responses to 5-ms tone pips were measured between subdermal electrodes (adjacent to the ipsilateral incision, at the vertex, and near the tail), amplified 10,000 times and filtered through a 0.3–3.0 kHz band-pass filter. For each frequency, the sound level starting below the threshold was increased in 5 dB steps and 512 responses. ABR thresholds were defined as the lowest level at which a repeatable waveform could be visually detected.

### Inferior colliculus recordings

Neural activity in the IC was recorded using a 16-channel, single-shank electrode array (177 $\mu m^2$/site, center-to-center electrode spacing of 50 µm; NeuroNexus Technologies, Ann Arbor, MI; A1 × 16-3 mm-50-177). Recordings were collected at a sampling rate of 25 kHz and analyzed offline using custom-written MATLAB scripts. Neighboring neurons in the IC have similar spectrotemporal preferences, and therefore multiple neurons generate action potentials at a similar timing (*Chen et al., 2012*). This often leads to poor isolation of single-unit activity in IC recordings (*Snyder et al., 2004*; *Rodríguez et al., 2010*; *Chen et al., 2012*; *Sadeghi et al., 2019*). In our recordings, distinguishable single-unit activities were observed from only a few channels, typically less than 3 while MUAs were evident in all channels (*Figure 2—figure supplement 1*). Distinguishable action potentials could be detected from only a few channels, typically less than 3. Thus, data analysis was based on MUAs, which were quantified by an analog representation of multiunit activity (aMUA). aMUA reflects the voltage signal power within the frequency range occupied by action potentials (*Chung et al., 1987*; *King and Carlile, 1994*; *Choi et al., 2010*; *Schnupp et al., 2015*, *Sadeghi et al., 2019*). This approach is advantageous over the more traditional measure of MUA based on thresholding and spike detection since aMUAs are not biased by free parameters (e.g., threshold levels), and provide a high signal-to-noise ratio (*Schnupp et al., 2015*). aMUA was measured as follows: (1) To extract MUA, the raw recordings were band-pass filtered between 325 and 6000 Hz (Butterworth IIR); during this process, low-frequency local field potentials and high-frequency electric noise signals were removed (*Figure 2—figure supplement 1A*). (2) The recorded signals prior to 2 ms after stimulus onset were excluded from subsequent analyses due to stimulus artifact (*Figure 2—figure supplement 1B*). (3) The extracted MUA signal was then rectified to take the absolute value of response magnitude over time. In addition, to reduce aliasing artifact, the processed signal was next low pass filtered at 475 Hz (Butterworth IIR), and downsampled from 25 to 12 kHz (*Figure 2—figure supplement 1C*). (4) The area under the curve for the time period 2–15 ms following stimulus onset was then calculated.

### *d*-Prime analysis

To quantify the change in neural responses to stimulus intensity, the discrimination index, *d′* (*d*-prime) was calculated by comparing aMUAs across successive pairs of stimulus levels in each electrode (*Macmillan and Creelman, 2004*; *Middlebrooks and Snyder, 2007*). Based on aMUAs to a given stimulus level and those to the next higher level, a *d′* value with unequal variance was calculated as

$$d^{'} = \left| \mu_a - \mu_b \right| / \sigma_{rms},$$

where $\mu$ and $\sigma_{rms}$ are the mean aMUA and common standard deviation, respectively (*Egan and Clarke, 1962*, *Das and Geisler, 2021*). The value of *d′* represents the distance between the means in units of a standard deviation – the larger the *d′* value, the more separated the distributions are. The *d′* values were then accumulated up to each stimulus intensity to calculate the cumulative *d′*.

## Construction of STC

To estimate the spread of activation from acoustic stimulation, previous studies measured the width of IC activation at an SPL of 10–40 dB above threshold. However, given that DRs are significantly different across modalities (e.g., the DRs of acoustic and electric stimulations are 25.96 ± 9.17 dB SPL and 3.24 ± 0.99 dB mA, respectively), comparing spatial spreads at a fixed dB level above threshold was not feasible. Alternatively, some studies measured spatial spreads at different dB levels above threshold for different modalities, for example 20 and 6 dB above threshold for acoustic and electric stimulation, respectively (*Snyder et al., 2004*). More recent studies that have evaluated novel stimulation modalities and compared them to acoustic and/or electric responses compared spatial spreads at a given response strength, typically at cumulative $d'$ values of 2–4 (*Middlebrooks and Snyder, 2007*; *Bierer et al., 2010*; *Moreno et al., 2011*; *Richter et al., 2011*; *George et al., 2015*; *Xu et al., 2019*; *Dieter et al., 2019*; *Keppeler et al., 2020*). Thus, to remain consistent with these previous studies, we also compared spectral spreads from acoustic, magnetic, and electric stimulation at cumulative discrimination indexes of 2 and 4. Based on the cumulative $d'$ values, an $n \times m$ matrix was constructed, where $n$ corresponded to stimulus intensity and $m$ to the electrode number. Iso-$d'$-contour-lines were derived by interpolating the matrix using MATLAB software. In all STCs (*Figures 2A–C and 4A–D*), contours for cumulative $d'$ levels of 1, 2, and 4 are shown. The stimulation threshold was selected as the cumulative $d'$ value of 1, and the best site (BS) was determined from the minima of the $d' = 1$ iso-contour. On average, the cumulative $d'$ levels of 2 and 4 correspond to 7.23 ± 5.34 and 18.53 ± 9.94 dB SPL above threshold for acoustic stimulation, 0.47 ± 0.30 and 1.41 ± 0.53 dB 1 mA above threshold for electric stimulation, and 2.57 ± 1.33 and 7.98 ± 5.41 dB 1 V above threshold for magnetic stimulation (*Figure 2—figure supplement 2*).

The spatial spread of the response was calculated as the distance between the ventral- and dorsal-most electrodes exhibiting a suprathreshold response ($d' > 1$). Using the characteristic frequency of each electrode obtained from the acoustic stimulation, the spatial spread could then be converted to the spectral spread of cochlear excitation, that is, the width of the corresponding frequency bands.

The spread of excitation was also evaluated by measuring the number of peaks observed in the ST profiles. The number of peaks was measured by the number of isolated electrode groups exhibiting a suprathreshold response ($d' > 1$) at the stimulus intensity that elicited a cumulative $d'$ value of 2.

## Statistical evaluation

All data were presented as the mean ± standard deviation. To verify statistical significance, a Student $t$-test was used for the data with a single independent variable (*Figure 6*) and two-way ANOVA with subsequent Tukey's test (GraphPad Prism 9.3.1) was used for those with multiple variables (*Figure 5*).

## Deafening

After recording acoustic responses, animals were acutely deafened by gently infusing 5 μl of distilled water through the round window to cause osmotic stress (*Chung et al., 2016*). Successful deafening was confirmed 10 min after injection by remeasuring ABR to acoustic stimuli – a sharp increase in ABR thresholds (≥75 dB SPL) or complete elimination of the waveform was considered evidence that hair cells were no longer functioning. Small amounts of residual ABR were attributed to acoustic crossover to the contralateral, nondeafened ear (*Harrison et al., 2013*).

To chronically deafen animals through lesions of hair cells, we used the ototoxic aminoglycoside antibiotic gentamicin. The round window niche was exposed in anesthetized animals, and a small piece of Gelfoam sponge soaked in 200 μg gentamicin in distilled water was applied to the round window membrane (*Heydt et al., 2004*). The skin was closed with sutures. The animal received postsurgical analgesia with meloxicam (2 mg/kg) and buprenorphine (0.05 mg/kg). Ten days after this procedure, the absence (or near absence) of ABR and DPOAE responses was used to confirm successful deafening; these animals were then used for electric and magnetic stimulation experiments.

## Cochlear whole mounts and confocal fluorescence immunohistochemistry

After completion of stimulation procedures, deeply anesthetized animals were intracardially perfused with 4% paraformaldehyde (PFA), both cochleae were extracted and processed as previously described (*Seist et al., 2020*). PFA was gently perfused through the round and oval windows. Cochleae were

postfixed for 2 hr in 4% PFA and decalcified in 0.12 M ethylenediaminetetraacetic acid for 72 hr. The decalcified spiraling cochleae were microdissected into four to six pieces, blocked with 5% normal horse serum and 0.3% Triton X-100 (TX-100) in PBS for 30 min at room temperature, and immunostained to label hair cells overnight at room temperature with rabbit anti-myosin 7A (1:200, #25-6790 Proteus Biosciences, Ramona, CA) diluted in 1% normal horse serum with 0.3% TX. After washing in PBS, cochlear pieces were incubated with Alexa Fluor 488-conjugated goat anti-rabbit antibody (#A-11008) and Alexa Fluor 647-conjugated phalloidin (#A22287) at 1:200 (Invitrogen, Carlsbad, CA) for 90 min. A cochlear frequency map was created by applying a custom ImageJ plug-in (available here) to images acquired at low magnification (×10 objective) on a fluorescent microscope (E800, Nikon, Melville, NY). Cochlear whole mounts were subsequently imaged with a confocal microscope (SP8, Leica, Wetzlar, Germany) using a ×63 glycerol-immersion objective (1.3 N.A.) at the 32-kHz cochlear frequency region. Inner hair cells per 100 µm were counted in the basal turn at 32, 45.25, and 64 kHz (*Landegger et al., 2019*).

## Acknowledgements

We thank Kenneth Hancock for his expert technical assistance.

## Additional information

### Funding

| Funder | Grant reference number | Author |
| --- | --- | --- |
| National Institutes of Health | DC 01089 | Stephen McInturff Daniel J Lee Christian Brown |
| Fondation Bertarelli | Translational Neuroscience and Neuro-Engineering | Stephen McInturff Daniel J Lee Christian Brown |
| National Institute on Deafness and Other Communication Disorders | R01 DC015824 | Konstantina M Stankovic Richard Seist |
| Fondation Bertarelli | Bertarelli Endowed Professorship | Konstantina M Stankovic |
| U.S. Department of Defense | CDMRP - VR170089 | Jae-Ik Lee Shelley Fried |
| BRAIN Initiative | NS110575 | Jae-Ik Lee Shelley Fried |
| Novo Nordisk Fonden | 0064289 | Jae-Ik Lee Shelley Fried |
| National Institutes of Health | R01- EY029022 | Jae-Ik Lee Shelley Fried |
| Remondi Foundation | | Konstantina M Stankovic |
| Larry Bowman | | Konstantina M Stankovic |

The funders had no role in study design, data collection, and interpretation, or the decision to submit the work for publication.

### Author contributions

Jae-Ik Lee, Richard Seist, Conceptualization, Data curation, Formal analysis, Investigation, Methodology, Software, Visualization, Writing – original draft; Stephen McInturff, Investigation, Methodology, Writing – review and editing; Daniel J Lee, Funding acquisition, Resources, Supervision, Writing – review and editing; M Christian Brown, Funding acquisition, Resources, Supervision, Validation, Writing – review and editing; Konstantina M Stankovic, Conceptualization, Funding acquisition, Methodology, Resources, Supervision, Validation, Writing – original draft; Shelley

Fried, Conceptualization, Funding acquisition, Investigation, Resources, Validation, Writing – original draft

### Author ORCIDs
Jae-Ik Lee http://orcid.org/0000-0002-9006-3405
Richard Seist http://orcid.org/0000-0003-1895-6513
Daniel J Lee http://orcid.org/0000-0001-5104-4626
M Christian Brown http://orcid.org/0000-0002-4420-5345
Konstantina M Stankovic http://orcid.org/0000-0003-0233-279X
Shelley Fried http://orcid.org/0000-0001-6456-8656

### Ethics
All procedures were approved by the Institutional Animal Care and Use Committee of Massachusetts Eye and Ear, and carried out in accordance with the NIH Guide for the Care and Use of Laboratory Animals (protocol # 15-003).

### Decision letter and Author response
Decision letter https://doi.org/10.7554/eLife.76682.sa1
Author response https://doi.org/10.7554/eLife.76682.sa2

## Additional files

### Supplementary files
• Transparent reporting form

### Data availability
The source data and codes are available on the Open Science Framework (https://doi.org/10.17605/OSF.IO/Y7ZRX).

The following dataset was generated:

| Author(s) | Year | Dataset title | Dataset URL | Database and Identifier |
|---|---|---|---|---|
| Jae-Ik L, Richard S, Stephen M, Daniel JL M, Christian B, Konstantina MS, Shelley F | 2022 | Source data for eLife76682 | https://doi.org/10.17605/OSF.IO/Y7ZRX | Open Science Framework, 10.17605/OSF.IO/Y7ZRX |

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
