## [Editor Report]

This study demonstrates that microcoil magnetic stimulation in the cochlea generates efficient and spatially precise activation of the auditory system. This new method opens interesting possibilities to improve the resolution of auditory restoration by implanted devices.

---

## [Decision Letter]

**Decision letter after peer review:**

Thank you for submitting your article "Magnetic Stimulation Allows Focal Activation of the Mouse Cochlea" for consideration by *eLife*. Your article has been reviewed by 3 peer reviewers, and the evaluation has been overseen by a Reviewing Editor and Barbara Shinn-Cunningham as the Senior Editor. The following individuals involved in the review of your submission have agreed to reveal their identity: Yann Nguyen (Reviewer #1); Tania Rinaldi Barkat (Reviewer #2).

Essential revisions:

1) Please answer all comments from the three referees, paying particular attention to the quantifications of the data. There are several figures which only show examples and thorough quantifications displaying the variability of the outcome should be added.

More specifically:

2) Quantification of the frequency response area FRA using the d' index is very puzzling should be complemented with more classical measures Q10dB, the Q40dB or the Octave distance which are classically used in auditory neuroscience.

3) Quantification of the spectral spread of activation used in figure 4A-B must be improved. Based on the 11 animals tested with ipsilateral tones (and not contralateral tones), the authors estimated that each electrode corresponds to a particular frequency, then the between-electrode distance is converted in an octave distance. This is an artificial, comparison of electrical and magnetic stimulation should be done based on electrode distance only.

4) By using ipsilateral sounds instead of contralateral sounds, the authors largely underestimated the acoustic inputs reaching the recording sites (because the main ascending pathways cross the midline between the cochlear nucleus and the superior olivary complex). Further experiments should be performed with contralateral recordings.

5). The Spatial Tuning Curves (STC) in the plots presented in Figure 1D,E,F are just classical Frequency Response Areas (FRA). FRA should be used instead of STC.

6) Supplementary figure 1 should be carefully improved. The quality of the data seems questionable given the amplitude of Action Potential (AP). Also, based on this figure the mean aMUA display for acoustic stimulation has a long onset latency and the response peaks after 10 ms. This type of response has nothing in common with the classic phasic responses or the phasic-tonic responses of IC neurons published from PSTHs over the last three decades (e.g. Langner and Schreiner 1988; Yin 1994; Condon et al. 1996; Palombi and Caspary 1996; Rees et al. 1997, Kuwada et al. 1997). The discrepancy should be discussed.

7) The claim that activation triggered by acoustic stimulation is narrow is questionable. It has been known for decades that although some IC neurons have V-shaped frequency response areas (FRA) similar to those of primary auditory nerve fibers, others have substantially different shapes indicative of the interplay of excitation and inhibition in shaping these receptive fields (Ehret and Merzenich 1988; Casseday and Covey 1992, Yang et al. 1992; Ramachandran et al. 1999; Hernandez et al. 2005; Palmer et al. 2013). Thus, the selective V-shaped shown in Figure 1 at presentation of pure tone stimuli is clearly not representative of the diversity of IC neurons. In addition, among the neurons exhibiting V-shaped FRA, many have responses extending in both directions 2-4 octaves away from the characteristic frequency (CF). The sharp tuning of pure tone responses in IC at 80dB (Figure 1D, E, F), is not fully compatible with the known physiological properties of these neurons. Similarly, the fact that the rate-level functions are monotonic as shown in Figure 1H is also totally incompatible with the known physiological properties of IC neurons (see for example from the large database analyzed in Palmer et al., 2013). More examples of IC neurons response diversity should be given and the quantification of frequency tuning should not depend on 'V-shapeness'.

*Reviewer #1 (Recommendations for the authors):*

In conclusion, I think that this is a very original and interesting article because it addresses a major problem of cochlear implantation with a disruptive approach.

If the comments in the Public Review could be answered, I will consider positively this work for a publication.

*Reviewer #2 (Recommendations for the authors):*

1) The introduction is really well written and has a lot of important and interesting information. It would, however, be helpful to also include a short paragraph on how the magnetic stimulation actually works. As *eLife* articles have a large and diverse readership, it would be important to have this information in the paper instead of only referring to other published papers. A schematic comparing magnetic and electrical stimulation, for example, added as a panel to figure 2 (maybe to illustrate the description of l.266-275), could also address this.

2) I am not sure what reason could explain that single action potentials could not be detected (l. 428). The recording electrodes have all the characteristics that should allow nice distinctions of action potentials. This should be discussed in more detail.

3) Some results are only described in the text and not presented in figures (l. 107, l. 172-176, l.205, l.227-235). I would suggest to illustrated all results in figures, and having all results in the Results section, including the controls described on l.227-235.

4) The mice used have a wide age range (6-16 weeks). Have you checked whether there is an age dependence on the results?

5) An important aspect of the success – or not – of cochlear implants is, in addition to spatial spread and dynamic range, the temporal precision of the signal. It would be interesting to compare the timing of responses between magnetic and electrical stimulation. If the temporal precision is as high with magnetic as with electrical stimulation, then I think this approach would definitely represent an exciting improvement to classical cochlear implants.

6) The results of Figure 4c are described differently in the Results section. It would be nice if the figure matches the description (as well as if it shows examples of peaks).

7) In figure 1G, only 3 out of 11 mice have responses to 8 kHz. Is this because the frequency was not tested, or because the recording electrode was not in the 8 kHz region of the IC? If this was the case, how could that (mis)placement influence the results and interpretations of the magnetic and electrical stimulation?

8) For Figure 4a, 4b, and 5c, a multiple comparison test would be more appropriate than a student t-test.

*Reviewer #3 (Recommendations for the authors):*

1. There are many basic notions regarding the auditory physiology that the authors should handle more carefully in the entire experiment.

1a. Throughout the entire manuscript, the authors are talking about Spatial Tuning Curves (STC) but there is absolutely no spatial selectivity tested here. Spatial receptive fields correspond to situations where a set of calibrated speakers are placed at different azimuths or different elevations to test the spatial selectivity to pure tones or broadband noises (e.g. Brugge et al. 1998, 2001; Yao et al. 2015). What is tested here is the spectral selectivity of IC neurons, and as it is tested throughout a range of intensity, the plots presented in figure 1D,E,F are just classical Frequency Response Areas (FRA) that have been classified for decades with robust and relevant parameters.

1.b. In this paper, the FRA limits/borders are defined by the d' index sets at 1, 2 or 4, which does not tell anything about the tuning breadth. The FRA is classically used to determine the range of frequencies/intensities triggering significant responses above spontaneous firing rate and to know what is the value of firing rate reach at each intensity. Comparing pairs of stimulus levels for each electrode (methods line 445-446 for the explanation of the d') has no meaning. Detecting a difference between the average Multi-unit Activity (aMUA) tell you neither if there is a significant response, nor what is the strength of the response.

1.c. In addition, the supplementary figure 1 that is supposed to display what represents the aMUA is very difficult to consider in a quality paper. I cannot recognize a Multi-Unit recording based on these traces because there is no Action Potential (AP) of correct amplitude visible on this figure. Based on this figure the mean aMUA display for acoustic stimulation has a long onset latency and the response peaks after 10 ms. This type of response has nothing in common with the classic phasic responses or the phasic-tonic responses of IC neurons published from PSTHs over the last three decades (e.g. Langner and Schreiner 1988; Yin 1994; Condon et al. 1996; Palombi and Caspary 1996; Rees et al. 1997, Kuwada et al. 1997).

1.d. The claim that activation triggered by acoustic stimulation is narrow is, at least, questionable. It has been know for decades that although some IC neurons have V-shaped frequency response areas (FRA) similar to those of primary auditory nerve fibers, others have substantially different shapes indicative of the interplay of excitation and inhibition in shaping these receptive fields (Ehret and Merzenich 1988; Casseday and Covey 1992, Yang et al. 1992; Ramachandran et al. 1999; Hernandez et al. 2005; Palmer et al. 2013). Thus, the selective V-shaped shown in Figure 1 at presentation of pure tone stimuli is clearly not representative of the diversity of IC neurons. In addition, among the neurons exhibiting V-shaped FRA, many have responses extending in both directions 2-4 octaves away from the characteristic frequency (CF). Thus, the idea conveyed in this paper that the tone response of IC neurons is narrow at 80dB (Figure 1D, E, F), is just incompatible with the known physiological properties of these neurons. Similarly, the fact that the rate-level functions are monotonic as shown in Figure 1H is also totally incompatible with the known physiological properties of IC neurons (see for example from the large database analyzed in Palmer et al., 2013).

---

## [Author Response]

Essential revisions:1) Please answer all comments from the three referees, paying particular attention to the quantifications of the data. There are several figures which only show examples and thorough quantifications displaying the variability of the outcome should be added.

We appreciate the comments from the reviewers as well as the editor’s feedback and summary. All comments have been addressed and a summary of our responses is in this document; changes are tracked in the revised document. We have performed additional analysis for data quantification and provide explanation as to our rationale for choosing the analysis methods we used. We have also added more figures for ABR and IC responses in the figure supplements (Figure 3 —figure supplement 1; Figure 4 —figure supplements 1 and 2) and presented data points for individual samples in each plot. All source data used to make figures have been uploaded to the repository following the guideline of the *eLife* journal. We believe this will help the readers assess our results quantitatively.

More specifically:2) Quantification of the frequency response area FRA using the d' index is very puzzling should be complemented with more classical measures Q10dB, the Q40dB or the Octave distance which are classically used in auditory neuroscience.

In general, we tried to use conventional methods so that readers can readily understand and interpret our results. We acknowledge that measuring the spread of activation at certain dB levels above threshold is commonly used to evaluate responses to acoustic stimulation. However, we felt that the use of the classic Q10dB or Q40dB measures to compare the spread of activation across different modalities was less suitable since each modality has a different dynamic range. For example, the dynamic range of electric stimulation was only ~ 3 dB, while that of acoustic stimulation was more than 20 dB. Therefore, we adopted an approach based on fixed significance of response strengths, i.e., measuring at an identical discrimination index. In this way, the estimation of the spread of excitation becomes independent of the stimulus’s nature and makes neural activation by different modalities more comparable. This approach is similar to that used in many previous studies in which new stimulation paradigms were evaluated (Middlebrooks et al., 2007; Bierer et al., 2010; Moreno et al., 2011; Richter et al., 2011; George et al., 2015; Xu et al., 2019; Dieter et al., 2019; Keppeler et al., 2020) and thus allows the performance of micro-coils to be more easily compared. Nevertheless, we agree that providing more detailed explanations would be helpful to many readers and have added additional language in the Method section of the revised manuscript.

[Line 528] “The value of d’ represents the distance between the means in units of a standard deviation – the larger the d’ value, the more separated the distributions are.”

[Line 532] “To estimate the spread of activation from acoustic stimulation, previous studies measured the width of IC activation at a sound pressure level of 10 – 40 dB above threshold. However, given that dynamic ranges are significantly different across modalities (e.g., the dynamic ranges of acoustic and electric stimulations are 25.96 ± 9.17 dB SPL and 3.24 ± 0.99 dB mA, respectively.), comparing spatial spreads at a fixed dB level above threshold was not feasible. Alternatively, some studies measured spatial spreads at different dB levels above threshold for different modalities., e.g., 20 dB and 6 dB above threshold for acoustic and electric stimulation, respectively (Snyder et al., 2004). More recent studies that have evaluated novel stimulation modalities and compared them to acoustic and/or electric responses compared spatial spreads at a given response strength, typically at cumulative d’ values of 2-4 (Middlebrooks and Snyder, 2007, Bierer et al., 2010, Moreno et al., 2011, Richter et al., 2011, George et al., 2015, Xu et al., 2019, Dieter et al., 2019, Keppeler et al., 2020). Thus, to remain consistent with these previous studies, we also compared spectral spreads from acoustic, magnetic, and electric stimulation at cumulative discrimination indexes of 2 and 4.”

We have also plotted the cumulative *d’* index with respect to dB levels above threshold for each modality (Figure 2 —figure supplement 2) and added relevant descriptions in the Materials and methods section. We believe these will facilitate understanding of our results by readers, especially those who are accustomed to the analysis based on fixed dB levels.

[Line 550] “On average, the cumulative d′ levels of 2 and 4 correspond to 7.23 ± 5.34 and 18.53 ± 9.94 dB SPL above threshold for acoustic stimulation, 0.47 ± 0.30 and 1.41 ± 0.53 dB 1 mA above threshold for electric stimulation, and 2.57 ± 1.33 and 7.98 ± 5.41 dB 1 V above threshold for magnetic stimulation (Figure 2 —figure supplement 2).”

3) Quantification of the spectral spread of activation used in figure 4A-B must be improved. Based on the 11 animals tested with ipsilateral tones (and not contralateral tones), the authors estimated that each electrode corresponds to a particular frequency, then the between-electrode distance is converted in an octave distance. This is an artifical, comparison of electrical and magnetic stimulation should be done based on electrode distance only.

As the reviewer mentioned, we converted the distance of activated electrodes to octave distance based on the characteristic frequency of each electrode derived from Figure 2D. This translation provides an estimate of the activated frequency band across the tonotopic organization of the cochlea by stimulation and previous studies evaluating novel methods of artificial stimulation presented the spread of activation by artificial stimulation in a similar way (Dieter et al., 2019; Keppeler et al., 2020). Therefore, we felt the use of this approach would provide the most direct comparison to previous work. Nevertheless, we agree that quantifying the activation spread by electrode distance would be more intuitive to some readers and have added the corresponding plots in Figure 5.

4) By using ipsilateral sounds instead of contralateral sounds, the authors largely underestimated the acoustic inputs reaching the recording sites (because the main ascending pathways cross the midline between the cochlear nucleus and the superior olivary complex). Further experiments should be performed with contralateral recordings.

We thank the reviewers for highlighting the anatomy of the auditory pathway and, specifically, its crossing over the midline. The wording in the original manuscript was confusing as all modes of stimulation (acoustic, electric, magnetic) were delivered to the left cochlea and responses measured from the right inferior colliculus (IC); the side to which stimulation was delivered was referred to as ipsilateral and the opposite side was referred to as contralateral. We revised the wording as shown below and believe it will greatly reduce the potential for confusion.

[Line 100] “We stimulated the left cochlea with acoustic, electric, and magnetic stimuli and measured responses from a 16-channel recording array implanted along the tonotopic axis of the right (contralateral) inferior colliculus (IC) in anesthetized mice (Figure 1C; Materials and Methods).”

5). The Spatial Tuning Curves (STC) in the plots presented in Figure 1D,E,F are just classical Frequency Response Areas (FRA). FRA should be used instead of STC.

IC responses can be presented by both frequency response areas (FRA) and spatial tuning curves (STC). FRA typically represents neuronal responses to different sound frequencies, recorded at a fixed site, while STC represents neuronal responses to acoustic stimulation with a fixed frequency or artificial stimulation at a fixed intracochlear location, recorded at multiple sites. FRA and STC constructed from the same data set are presented in Snyder et al., 2008. Although the patterns of FRA and STC are often similar in appearance, they serve different purposes; FRA conveys information about the excitatory/inhibitory circuits of auditory pathways and frequency tuning functions while STC visualizes the activity evoked across the tonotopic organization by stimulation at a fixed site (or fixed sound frequency). Because one of the main goals of our study was to compare the spread of activation across different stimulation modalities, we used STC, as was done in many previous studies (Middlebrooks et al., 2007; Snyder et al., 2008; Bierer et al., 2010; Moreno et al., 2011; Richter et al., 2011; George et al., 2015; Xu et al., 2019; Dieter et al., 2019; Keppeler et al., 2020; Thompson et al., 2020).

6) Supplementary figure 1 should be carefully improved. The quality of the data seems questionable given the amplitude of Action Potential (AP). Also, based on this figure the mean aMUA display for acoustic stimulation has a long onset latency and the response peaks after 10 ms. This type of response has nothing in common with the classic phasic responses or the phasic-tonic responses of IC neurons published from PSTHs over the last three decades (e.g. Langner and Schreiner 1988; Yin 1994; Condon et al., 1996; Palombi and Caspary 1996; Rees et al., 1997, Kuwada et al., 1997). The discrepancy should be discussed.

We acknowledge that there was a mistake in the amplitude unit and have corrected this in the revised manuscript (µV mV); we thank the Reviewers for their attention to detail. We suspect the latency differences arise from interspecies variability, i.e., previous studies that report peak onsets of 8-15 ms were not performed in mice. We point out that one previous study in mice (Land et al., 2016) shows a peak latency of ~10 ms, similar to what we observe here. We again thank the reviewer for making us aware of reporting the incorrect amplitude.

7) The claim that activation triggered by acoustic stimulation is narrow is questionable. It has been known for decades that although some IC neurons have V-shaped frequency response areas (FRA) similar to those of primary auditory nerve fibers, others have substantially different shapes indicative of the interplay of excitation and inhibition in shaping these receptive fields (Ehret and Merzenich 1988; Casseday and Covey 1992, Yang et al., 1992; Ramachandran et al., 1999; Hernandez et al., 2005; Palmer et al., 2013). Thus, the selective V-shaped shown in Figure 1 at presentation of pure tone stimuli is clearly not representative of the diversity of IC neurons. In addition, among the neurons exhibiting V-shaped FRA, many have responses extending in both directions 2-4 octaves away from the characteristic frequency (CF). The sharp tuning of pure tone responses in IC at 80dB (Figure 1D, E, F), is not fully compatible with the known physiological properties of these neurons. Similarly, the fact that the rate-level functions are monotonic as shown in Figure 1H is also totally incompatible with the known physiological properties of IC neurons (see for example from the large database analyzed in Palmer et al., 2013). More examples of IC neurons response diversity should be given and the quantification of frequency tuning should not depend on 'V-shapeness'.

In the studies mentioned by the reviewer, IC single-units were classified as “V”, “I” (narrow tuning at all suprathreshold levels), “O” (narrow tuning at threshold and no responses at suprathreshold) as well as other less-frequent types. All of these studies report single-unit FRAs to acoustic stimulation and neurons with “V-shaped” tuning curves were reported as the most abundant (Casseday and Covey 1992). In the present study, however, we recorded multiunit responses; combining responses from multiple types of single-units, as would occur with our multiunit recordings, are dominated by neurons with “V-shaped” with lesser contributions from other types, and almost certainly results in a V-shaped tuning curve (a good example of this is Figure 2 in Palmer et al., 2013 and Figure 3 in Snyder et al., 2008). In addition, Figures 1D-F in the original figure set showed spatial tuning curves (STCs), not FRAs. The difference between STC and FRA is described in the answer for essential review Q5. The STCs based on IC multi-unit activities typically have narrow V-shaped curves (Figure 4 in Snyder et al., 2008; Figure 1 in Dieter et al., 2019). Thus, while we agree that (original) Figures 1D-F do not appear to be representative of single unit FRAs, we think they are representative of multi-unit STCs. We have added language to clarify this issue as follows:

[Line 108] “The MUAs, which are combined responses from single-unit activities with various tuning curves, typically have “V” shaped tuning curves (Snyder et al., 2008) and MUA-based STCs for acoustic stimulation also have narrow “V” shaped curves.”

Reviewer #2 (Recommendations for the authors):1) The introduction is really well written and has a lot of important and interesting information. It would, however, be helpful to also include a short paragraph on how the magnetic stimulation actually works. As eLife articles have a large and diverse readership, it would be important to have this information in the paper instead of only referring to other published papers. A schematic comparing magnetic and electrical stimulation, for example, added as a panel to figure 2 (maybe to illustrate the description of l.266-275), could also address this.

Thank you for the valuable suggestion. We added the following illustration in Figure 1.

2) I am not sure what reason could explain that single action potentials could not be detected (l. 428). The recording electrodes have all the characteristics that should allow nice distinctions of action potentials. This should be discussed in more detail.

This is also a valid point, and the following language has been added.

[Line 497] “Neighboring neurons in the IC have similar spectro-temporal preferences, and therefore multiple neurons generate action potentials at a similar timing (Chen et al. 2012). This often leads to poor isolation of single-unit activity in IC recordings (Snyder et al., 2004; Rodríguez et al., 2010; Chen et al. 2012; Sadeghi et al., 2019).”

3) Some results are only described in the text and not presented in figures (l. 107, l. 172-176, l.205, l.227-235). I would suggest to illustrated all results in figures, and having all results in the Results section, including the controls described on l.227-235.

Thanks for alerting us to this; appropriate modification has been made as follows.

[Line 107 (original manuscript)/121 (resubmitted manuscript)] We have indicated that the relevant data is shown in Figure 6D.

[Line 172-176/186-190] We have added all STCs in the supplementary document (Figure 4 —figure supplements 1 and 2).

[Line 205/219] We used ABR and DPOAE as an “on-screen” evaluation for successful deafening. DPOAE amplitudes were completely flat. We have included more ABR data in the supplementary figure.

[Line 227-233/247-251] We did not make a record for the impedance of insulation or the DC resistance of the coil leads. Instead, once a coil failed (i.e., the impedance of insulation < 200 MΩ or DC resistance of the coil leads > 10 Ω), the coil was discarded and recordings obtained from the coil were exempted from the analysis. We clarified this in the text.

[Line 247]“If the impedance of the coil insulation was found to be less than 200 MΩ after an experiment, recordings obtained from the coil were exempted from the analysis.”

[Line 250] “The coils with DC resistance over 10 Ω were discarded.”

[Line 233-235/251-254] The temperature of the coils was measured in previous studies. We cited the relevant studies in the manuscript.

4) The mice used have a wide age range (6-16 weeks). Have you checked whether there is an age dependence on the results?

We intended to use young adult mice with good hearing (as evaluated by ABR and DPOAE thresholds) prior to deafening. In our cohort, we did not observe any differences attributable to age.

5) An important aspect of the success – or not – of cochlear implants is, in addition to spatial spread and dynamic range, the temporal precision of the signal. It would be interesting to compare the timing of responses between magnetic and electrical stimulation. If the temporal precision is as high with magnetic as with electrical stimulation, then I think this approach would definitely represent an exciting improvement to classical cochlear implants.

This is an important point. As the reviewer mentioned, one of the important goals of artificial stimulation is to evoke high rate auditory responses with high temporal precision. SGNs can follow a rate of auditory stimulation up to a few hundred Hz and maintain sub-millisecond precision of spike timing, which is crucial for auditory function (Heil and Peterson, 2015). In this study, we delivered electric and magnetic stimulation with a pulse rate of 25 Hz. Although there was no significant change in response probability and timing across pulses, this pulse rate is too low to assess temporal fidelity. We are very interested in the temporal kinetics of auditory responses to magnetic stimulation and will report our findings in future studies.

6) The results of Figure 4c are described differently in the Results section. It would be nice if the figure matches the description (as well as if it shows examples of peaks).

We thank the reviewer for pointing this out. We have added the number of STCs in each group in Figure 5E so that the figure matches the description.

7) In figure 1G, only 3 out of 11 mice have responses to 8 kHz. Is this because the frequency was not tested, or because the recording electrode was not in the 8 kHz region of the IC? If this was the case, how could that (mis)placement influence the results and interpretations of the magnetic and electrical stimulation?

Responses to 8 kHz were tested in every animal. The fact that only 3 mice have responses to 8 kHz likely results from the positioning of the recording probe, i.e., positioning missed the IC region that responds best to that frequency. The 8 kHz is a fairly low characteristic frequency that would be expected to be represented toward the apex of the cochlea. Nevertheless, responses to electric and magnetic stimulation always had the best electrodes clearly within the range of the recording probe (Figures. 4B and D). Thus, we consider this issue to be a minor one, but added the following sentence to the Results.

[Line 111] “Only 5 out of 11 animals had best sites at 8 kHz (Figure 2D) and thus the sampling of the lowest-frequency region of the IC may have been incomplete”.

8) For Figure 4a, 4b, and 5c, a multiple comparison test would be more appropriate than a student t-test.

Thank you for alerting us to this. We have analyzed the data with multiple independent variables with ANOVA (e.g., Figures 5A-D) while maintaining a student t-test for the data with a single independent variable.

Reviewer #3 (Recommendations for the authors):1. There are many basic notions regarding the auditory physiology that the authors should handle more carefully in the entire experiment.1a. Throughout the entire manuscript, the authors are talking about Spatial Tuning Curves (STC) but there is absolutely no spatial selectivity tested here. Spatial receptive fields correspond to situations where a set of calibrated speakers are placed at different azimuths or different elevations to test the spatial selectivity to pure tones or broadband noises (e.g. Brugge et al., 1998, 2001; Yao et al., 2015). What is tested here is the spectral selectivity of IC neurons, and as it is tested throughout a range of intensity, the plots presented in figure 1D,E,F are just classical Frequency Response Areas (FRA) that have been classified for decades with robust and relevant parameters.

This comment is addressed in the Essential revisions section (Q5).

1.b. In this paper, the FRA limits/borders are defined by the d' index sets at 1, 2 or 4, which does not tell anything about the tuning breadth. The FRA is classically used to determine the range of frequencies/intensities triggering significant responses above spontaneous firing rate and to know what is the value of firing rate reach at each intensity. Comparing pairs of stimulus levels for each electrode (methods line 445-446 for the explanation of the d') has no meaning. Detecting a difference between the average Multi-unit Activity (aMUA) tell you neither if there is a significant response, nor what is the strength of the response.

The difference between FRA and STC is explained in the Essential revisions section (Q5).

As the reviewer understood, the *d’* value was calculated by comparing aMUAs across successive pairs of stimulus levels, but then accumulated up to each stimulus intensity to calculate the cumulative *d’* value. The cumulative d’ value essentially provides a comparison between responses to each stimulus level and spontaneous activities. Thus, the contour lines based on the cumulative *d’* index indicate the range of IC electrodes and stimulus intensities where significant responses above spontaneous activity were observed. Our approach is also consistent with many previous studies which assessed the spread of activation of new stimulation strategies. Nevertheless, we thought that it would be helpful to provide more detailed explanations. Please see the answer for the Essential revisions Q2.

1.c. In addition, the supplementary figure 1 that is supposed to display what represents the aMUA is very difficult to consider in a quality paper. I cannot recognize a Multi-Unit recording based on these traces because there is no Action Potential (AP) of correct amplitude visible on this figure. Based on this figure the mean aMUA display for acoustic stimulation has a long onset latency and the response peaks after 10 ms. This type of response has nothing in common with the classic phasic responses or the phasic-tonic responses of IC neurons published from PSTHs over the last three decades (e.g. Langner and Schreiner 1988; Yin 1994; Condon et al., 1996; Palombi and Caspary 1996; Rees et al., 1997, Kuwada et al., 1997).

This comment is addressed in the Essential revisions section (Q6).

1.d. The claim that activation triggered by acoustic stimulation is narrow is, at least, questionable. It has been know for decades that although some IC neurons have V-shaped frequency response areas (FRA) similar to those of primary auditory nerve fibers, others have substantially different shapes indicative of the interplay of excitation and inhibition in shaping these receptive fields (Ehret and Merzenich 1988; Casseday and Covey 1992, Yang et al., 1992; Ramachandran et al., 1999; Hernandez et al., 2005; Palmer et al., 2013). Thus, the selective V-shaped shown in Figure 1 at presentation of pure tone stimuli is clearly not representative of the diversity of IC neurons. In addition, among the neurons exhibiting V-shaped FRA, many have responses extending in both directions 2-4 octaves away from the characteristic frequency (CF). Thus, the idea conveyed in this paper that the tone response of IC neurons is narrow at 80dB (Figure 1D, E, F), is just incompatible with the known physiological properties of these neurons. Similarly, the fact that the rate-level functions are monotonic as shown in Figure 1H is also totally incompatible with the known physiological properties of IC neurons (see for example from the large database analyzed in Palmer et al., 2013).

This comment is addressed in the Essential revisions section (Q7).